# Cyclic di-GMP differentially tunes a bacterial flagellar motor through a novel class of CheY-like regulators

Jutta Nesper[1†‡], Isabelle Hug[1†], Setsu Kato[2†], Chee-Seng Hee[3†], Judith Maria Habazettl[3], Pablo Manfredi[1], Stephan Grzesiek[3], Tilman Schirmer[3], Thierry Emonet[2,4], Urs Jenal[1]*

[1]Focal Area of Infection Biology, Biozentrum of the University of Basel, Basel, Switzerland; [2]Department of Molecular, Cellular and Developmental Biology, Yale University, New Haven, United States; [3]Focal Area of Structural Biology and Biophysics, Biozentrum of the University of Basel, Basel, Switzerland; [4]Department of Physics, Yale University, New Haven, United States

**Abstract** The flagellar motor is a sophisticated rotary machine facilitating locomotion and signal transduction. Owing to its important role in bacterial behavior, its assembly and activity are tightly regulated. For example, chemotaxis relies on a sensory pathway coupling chemical information to rotational bias of the motor through phosphorylation of the motor switch protein CheY. Using a chemical proteomics approach, we identified a novel family of CheY-like (Cle) proteins in *Caulobacter crescentus*, which tune flagellar activity in response to binding of the second messenger c-di-GMP to a C-terminal extension. In their c-di-GMP bound conformation Cle proteins interact with the flagellar switch to control motor activity. We show that individual Cle proteins have adopted discrete cellular functions by interfering with chemotaxis and by promoting rapid surface attachment of motile cells. This study broadens the regulatory versatility of bacterial motors and unfolds mechanisms that tie motor activity to mechanical cues and bacterial surface adaptation.

DOI: https://doi.org/10.7554/eLife.28842.001

*For correspondence: urs.jenal@
unibas.ch

†These authors contributed
equally to this work

‡Deceased

**Competing interests:** The
authors declare that no
competing interests exist.

**Reviewing editor:** Tâm Mignot,
Aix Marseille University-CNRS
UMR7283, France

## Introduction

In their natural habitats bacteria are exposed to rapidly changing environmental conditions and need to find optimal requirements for growth and survival. To explore new habitats, many bacteria move towards favorable places or colonize liquid-surface interfaces. A powerful device facilitating such behavior is the flagellar motor, a complex rotary organelle used for cell dispersal and as sensory apparatus to communicate mechanical signals upon surface encounter (*Berg, 2008*; *Harshey and Partridge, 2015*). Given their central role in bacterial cell behavior and virulence it is not surprising that bacteria tightly control assembly and activity of flagella (*Chevance and Hughes, 2008*; *Hazelbauer et al., 2008*; *Boehm et al., 2010*; *Paul et al., 2010*; *Mukherjee and Kearns, 2014*). Modulating motor activity allows bacteria to change their swimming trajectories and by that respond to changes in chemical composition, light intensity or temperature and move towards more favorable regions. Flagellar motors rotate either clockwise (CW) or counterclockwise (CCW) and by adjusting reversal frequencies, bacteria travel along gradients of favored or repulsing conditions (*Berg, 2008*; *Hazelbauer et al., 2008*).

The best-understood taxis system is the chemotaxis pathway of *E. coli*. This organism senses the external concentration of sugars or amino acids via large clusters of membrane-bound chemoreceptors. This information is relayed to a chemoreceptor coupled histidine kinase, CheA, which in turn

phosphorylates a diffusible response regulator, CheY. CheY ~P interacts with the flagellar switch proteins FliM and FliN to redirect motor rotation from CCW to CW (*Welch et al., 1993*; *Sarkar et al., 2010*). This prompts cells to change from moving forward in straight runs to tumbling, thereby randomizing their courses. By adjusting the tumbling frequency to the external milieu, this simple signal transduction pathway directs cellular movement in chemical gradients (*Krell et al., 2011*; *Porter et al., 2011*; *Scharf et al., 1998*). Many bacteria harbor chemotaxis systems that far exceed the complexity of the canonical *E. coli* system. For example, *Rhodobacter spheriodes* has two distinct chemoreceptor clusters, one located in the membrane and one in the cytosol. Four different CheA kinases and six CheY proteins relay information from the two receptor clusters to two distinct flagellar motors (*Porter et al., 2011*). In this organism several CheYs operate in parallel and functionally interact to determine chemotactic behavior (*Ferré et al., 2004*; *Porter et al., 2006*; *Porter et al., 2008*; *Tindall et al., 2010*). Moreover, the interaction between CheY and flagellar motors has diversified during evolution to generate different behavioral responses. Although most CheYs induce motor reversal, individual CheY components can act as a molecular break to provoke motor stop or slow down motor rotation (*Porter et al., 2011*; *Pilizota et al., 2009*; *Attmannspacher et al., 2005*).

Recently, the second messenger c-di-GMP was shown to interfere with motor function and bacterial cell motility. In *E. coli* increased c-di-GMP levels result in dynamic modulation of motor torque by the c-di-GMP effector protein YcgR, which in its c-di-GMP-bound form interacts with the flagellar rotor/stator interface (*Boehm et al., 2010*). A similar mechanism was proposed to tune motility in *Bacillus subtilis* and in *Pseudomonas aeruginosa* (*Chen et al., 2012*; *Baker et al., 2016*). In the latter, the YcgR homolog FlgZ controls swarming motility by specifically interacting with the MotCD stator, which is required for swarming motility on surfaces. YcgR and its homologs adjust motor speed, a level of control that may promote the transition from a motile to a sessile lifestyle and help bacteria to colonize surfaces (*Boehm et al., 2010*; *Chen et al., 2012*). Additionally, c-di-GMP can interfere with chemotaxis. YcgR was proposed to also bind to the flagellar switch thereby imposing a CCW rotational bias in *E. coli* (*Paul et al., 2010*; *Fang and Gomelsky, 2010*). Likewise, c-di-GMP adjusts motor switching frequency by interfering with chemoreceptor signaling, either directly (*Russell et al., 2013*) or indirectly by adjusting the methylation state of MCPs (*Xu et al., 2016*).

In addition to their prominent role in bacterial taxis, flagella serve as mechanosensitive devices to help planktonic bacteria to interact with surfaces (*Harshey and Partridge, 2015*; *Belas, 2014*). Upon surface contact, many bacteria rapidly adapt their behavior by secreting adhesins or by inducing surface motility and virulence mechanisms. In *Vibrio parahaemolyticus* slowing the rotation of the polar flagellum on surfaces triggers the synthesis of hundreds of lateral flagella used for surface swarming (*Kawagishi et al., 1996*). Similarly, jamming the *C. crescentus* polar flagellum upon surface encounter was proposed to provoke instant production of holdfast, an exopolysaccharide glue, and to irreversibly anchor cells to the surface (*Li et al., 2012*; *Hoffman et al., 2015*). This was confirmed recently by a study demonstrating that motor interference during *C. crescentus* surface contact leads to the production of c-di-GMP and to subsequent allosteric activation of the holdfast machinery (*Hug et al., 2017*). In some bacteria, components of the chemotaxis system contribute to surface adaptation by the flagellum (*Harshey, 2003*). For instance, *E. coli* and *Salmonella che* mutants fail to swarm on surfaces, while mutations in the flagellar switch apparatus restore swarming (*Burkart et al., 1998*; *Wang et al., 2005*).

Here we analyze motility and surface behavior of *Caulobacter crescentus*, an aquatic α-proteobacterium with a dimorphic life cycle that produces a motile swarmer (SW) and a sessile stalked (ST) cell upon division (*Figure 1a*). ST cells adhere to surfaces via an adhesive holdfast, which is located at the tip of an extension of their cell body, the stalk. When ST cells divide, they assemble a single flagellum, chemoreceptors and surface adherent pili at the pole opposite the stalk. Accordingly, SW offspring are motile, perform chemotaxis and can use their elongated pili to eventually re-attach to surfaces. One of the principal regulators of *C. crescentus* polarity and cell fate determination is c-di-GMP (*Abel et al., 2013*; *Jenal et al., 2017*). Concentrations of c-di-GMP oscillate during the *C. crescentus* life cycle thereby promoting cell cycle progression and coordinating the assembly of cell type-specific polar organelles (*Abel et al., 2013*; *Lori et al., 2015*; *Davis et al., 2013*). In addition, c-di-GMP was shown to control flagellar motor activity (*Abel et al., 2013*; *Christen et al., 2007*), although the molecular mechanisms of this behavior are unclear.

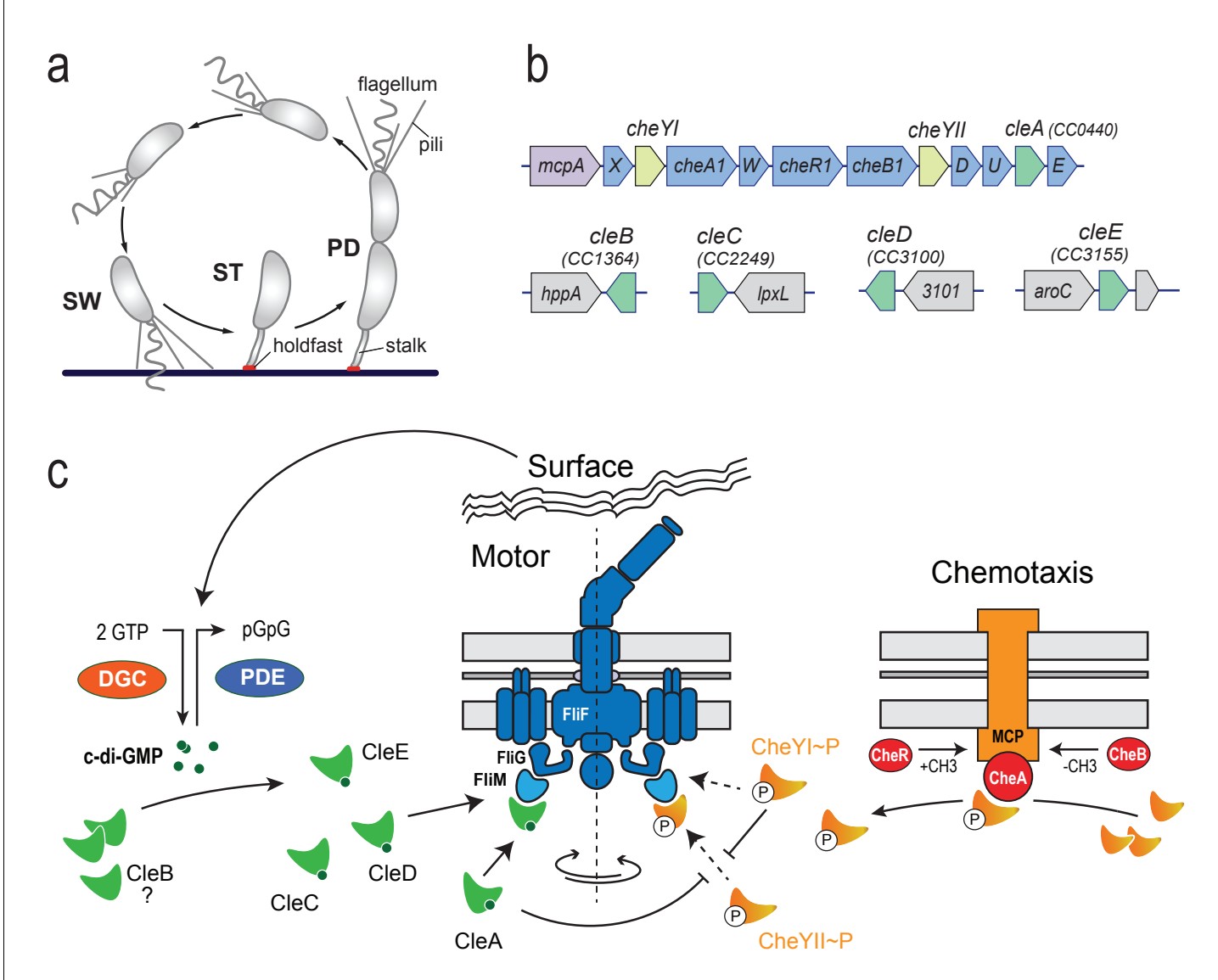

**Figure 1.** Cle proteins constitute a novel family of CheY like proteins in *C. crescentus*. (**a**) Schematic of the *C. crescentus* cell cycle. SW, swarmer; ST, stalked; PD, predivisional cell. (**b**) Genomic organization of the *C. crescentus* chemotaxis gene cluster and of regions containing *cle* genes. Purple: *mcp* gene (methyl-accepting chemotaxis protein); blue: *che* genes; light green: *cheY* genes; dark green *cle* genes; grey: hypothetical genes. (**c**) Model for CheY and Cle-mediated control of the *C. crescentus* flagellum. CheY (orange) and Cle proteins (green) interact with the flagellar switch protein FliM. CheYII is the functional homolog of *E. coli* CheY and is responsible to induce motor reversals upon phosphorylation by chemoreceptor-coupled kinases CheA (red) and subsequent binding to the FliM switch protein. CleA (in its c-di-GMP bound form) and CheYI ~P compete with CheYII ~P for FliM binding sites thereby reducing the overall reversal rate of the flagellar motor. Accordingly, CleA may promote smooth forward or backward swimming through CheYII competition under conditions that boost c-di-GMP levels in motile *C. crescentus* SW cells. Enzymes regulating the internal concentration of c-di-GMP (green) (DGC, red; PDE, blue) are indicated. CleC, CleE and possibly CleD interfere with motor function to boost rapid surface attachment. These proteins may be part of a feedback mechanism through which the flagellum signals surface encounter via DGCs and/or PDEs to increase the internal c-di-GMP pool (*Hug et al., 2017*).

DOI: https://doi.org/10.7554/eLife.28842.002

The following figure supplement is available for figure 1:

**Figure supplement 1.** Grouping and alignment of *C. crescentus* Cle proteins.

DOI: https://doi.org/10.7554/eLife.28842.003

*C. crescentus* SW cells display a three-step 'forward, reverse, and flick' swimming pattern (*Liu et al., 2014*). When rotating CW, the flagellum pushes the cell body forward tracing out a helical trajectory. Upon reversal to CCW rotation the flagellum pulls the cell backwards in a straight line (*Liu et al., 2014*; *Koyasu and Shirakihara, 1984*). A flick-like event re-orients cells when they change from CCW to CW, thereby randomizing cell movements (*Liu et al., 2014*; *Xie et al., 2011*). *C. crescentus* reversal rates are controlled by the chemotaxis machinery but the exact wiring of the respective regulatory components is not well understood (*Ely et al., 1986*; *Skerker et al., 2005*). *C. crescentus* has 19 chemoreceptors, two CheAs and 12 annotated CheYs. Many of the chemotaxis genes are arranged in two large *che* clusters (*Nierman et al., 2001*) and are co-transcribed with flagellar genes in the late predivisional cell (*Lasker et al., 2016*). Here we identify a novel class of CheY-like proteins in *C. crescentus* that do not receive signal input from chemoreceptor coupled CheA kinases but instead are activated by binding of c-di-GMP to short conserved C-terminal peptide sequences. We demonstrate that Cle proteins (C̲heY-l̲ike c-di-GMP e̲ffectors) bind to the canonical CheY binding site on the flagellar motor switch to adjust motor behavior. Despite their conserved interaction with the motor, individual Cle proteins have functionally diversified to control distinct cellular processes. These results add another layer of control to bacterial flagella revealing a complex regulatory network adjusting motor activity.

## Results

### Identification of a CheY subfamily as novel c-di-GMP effectors

Using c-di-GMP specific Capture Compound (cdG-CC) in combination with mass-spectrometry (CCMS) (*Nesper et al., 2012*; *Laventie et al., 2015*) we isolated three CheY-like response regulators, CC1364, CC2249 and CC3100, from *C. crescentus* cell extracts (*Table 1*). A phylogenetic analysis of all *C. crescentus* receiver domains revealed that these proteins group together with two additional CheY-like proteins (CC0440, CC3155) in a branch of the CheY/CheB related chemotaxis cluster (*Figure 1—figure supplement 1a*). This subcluster was classified before based on short C-terminal extensions, which distinguish them from canonical CheY proteins (*Galperin, 2004*). This extension can be subdivided into a conserved arginine-rich region (ARR) and a C-terminal region (CTR) of variable length and sequence identity (*Figure 1—figure supplement 1b*). Below we present evidence that members of this family specifically bind c-di-GMP and that this second messenger controls their activity. We thus renamed these proteins C̲heY-l̲ike c-di-GMP e̲ffectors CleA (CC0440), CleB (CC1364), CleC (CC2249), CleD (CC3100) and CleE (CC3155). While the *cleA* gene is part of the *C. crescentus* chemotaxis operon *che1*, other *cle* genes do not cluster with chemotaxis genes (*Figure 1b*).

To validate the CCMS results, we attempted to analyze c-di-GMP binding to purified Cle proteins. Unfortunately, only CleD was amenable to biochemical analyses while purification attempts with the other four members of this family produced insoluble protein. UV cross-linking experiments with purified Strep-tagged CleD and radiolabeled c-di-GMP revealed high affinity binding in the nanomolar range (*Figure 2a*). High affinity binding was confirmed by isothermal titration calorimetry (ITC) experiments revealing a $K_d$ of 86 nM (*Figure 2b*). CleD binds c-di-GMP specifically as unlabeled

**Table 1.** CheY-like response regulators identified by CCMS.

| Protein name | ID | CCMS experiment/CCMS competition* | | | |
|---|---|---|---|---|---|
| | | #spectral counts of identified peptides | | | |
| Experiment[†] | | 1 | 2 | 3 | 4 |
| CleB | CC1364 | 2/0 | 3/0 | 1/0 | 1/0 |
| CleC | CC2249 | 2/0 | 5/0 | 3/0 | 4/2 |
| CleD | CC3100 | 2/0 | 4/0 | 6/0 | 4/0 |

*All competition experiments were performed in the presence of 1 mM c-di-GMP.

[†]Experiment 1 was performed with 10 µM cdG-CC, experiment 2 with 10 µM cdG-CC, experiment 3 with 5 µM cdG-CC, and experiment 4 with 2.5 µM cdG-CC.

DOI: https://doi.org/10.7554/eLife.28842.004

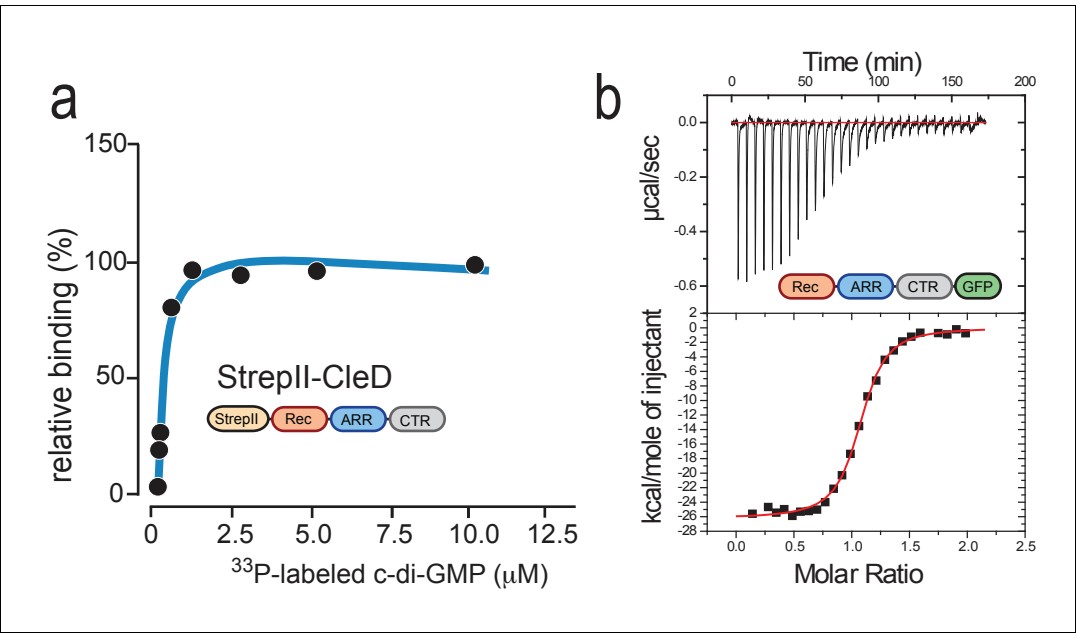

**Figure 2.** CleD specifically binds c-di-GMP. (**a**) UV-cross-linking of purified StrepII-CleD (1 μM) with increasing amounts of $^{33}$P-labelled c-di-GMP. The $K_d$ was determined to 212 nM. (**b**) ITC experiment analyzing c-di-GMP binding to CleD-GFP. The top panel shows the raw ITC data collected at 10°C, the bottom panel depicts the integrated titration peaks. The $K_d$ for c-di-GMP binding to CleD-GFP was calculated as 86 nM. Experiments in (**a**) and (**b**) include two technological replicas with a representative example shown.
DOI: https://doi.org/10.7554/eLife.28842.005

The following figure supplement is available for figure 2:

**Figure supplement 1.** Biochemical and biophysical characterization of CleD.
DOI: https://doi.org/10.7554/eLife.28842.006

c-di-GMP but not related nucleotides were able to effectively compete with the labeled probe for binding (*Figure 2—figure supplement 1a*). Specific binding was confirmed when using a more sensitive NMR approach (*Habazettl et al., 2011*). Because the imino protons H1 of free guanosines rapidly exchange with water but are involved in hydrogen bonds when bound to protein, their signal can only be observed when c-di-GMP is bound to the effector. When recording the difference spectrum of CleD with and without c-di-GMP, the four imino proton H1 resonances were found far downfield shifted (between 14ppm and 10.5ppm), indicating that two molecules of c-di-GMP are bound to one CleD protomer (*Figure 2—figure supplement 1b*). Finally, multi-angle light scattering (MALS) experiments indicated that the CleD protein is a monomer both in its apo- and ligand-bound forms (*Figure 2—figure supplement 1c*). Altogether, these data demonstrated that CleD is a member of a novel family of CheY-like proteins that bind c-di-GMP with high affinity and specificity.

## An Arg-rich peptide is required and sufficient for c-di-GMP binding of Cle proteins

We next used microscale thermophoresis (MST) to compare the binding of purified CleD variants with different C-terminal truncations to 2'fluoro-AHC-c-di-GMP (flr-cdG). The $K_d$ values of full-length CleD and a variant lacking part of the CTR were determined as 1.2 μM and 1.3 μM, respectively (*Figure 3a*). These values are roughly 10-fold higher as compared to the affinities determined for c-di-GMP itself, reflecting binding interference of the fluorescein group of flr-cdG. The dissociation constant of the Rec-ARR variant was 5-fold higher (6.2 μM), while the dissociation constant of the Rec domain alone was increased by 40-fold (41.7 μM) as compared to the full-length protein (*Figure 3a*). To test if the ARR of CleD is sufficient for c-di-GMP binding we fused it to the C-terminus of the *E. coli* CheY protein. While purified *E. coli* CheY protein failed to bind flr-cdG, the chimeric EcCheY-ARR protein bound flr-cdG with a $K_d$ of 14.8 μM (*Figure 3b*). ITC experiments with

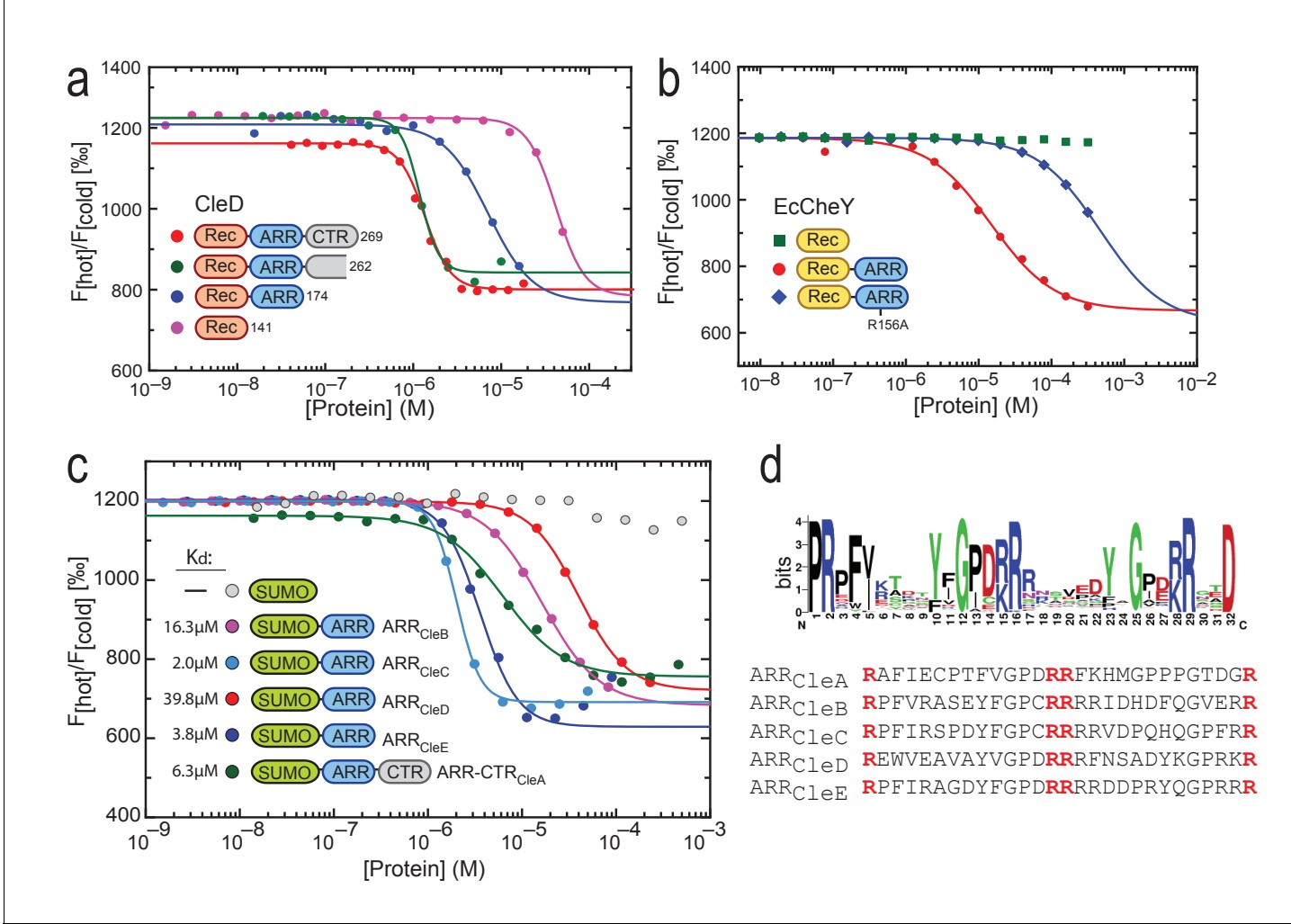

**Figure 3.** C-di-GMP binds to C-terminal Arg-rich peptides. (**a**) c-di-GMP binding to CleD requires the ARR. Microscale thermophoresis (MST) experiments with purified full-length His-CleD and truncated variants using 60 nM of the ligand 2'-Fluo-AHC-c-di-GMP (flr-cdG). $K_d$s were determined as follows: 1.2 μM for His-CleD (aa 19–269), 1.3 μM for His-CleD (aa 19–262), 6.2 μM for His-CleD-rec-ARR (aa 19–174) and 41.7 μM for His-CleD-rec (aa 19–141). (**b**) The ARR peptide is sufficient to bind c-di-GMP. The ARR of CleD was fused to the C-terminus of *E. coli* CheY. Binding of flr-cdG to purified His-CheY (green) and its variants containing a wild-type ARR (red) or the R156A mutant ARR (blue) was determined by MST. Apparent $K_d$s were calculated as 14.8 μM (red) and 471 μM (blue). (**c**) c-di-GMP binds to the ARR peptides of all five Cle proteins. MST experiments were done as outlined above. The ARR peptide regions fused to His-SUMO are indicated in *Figure 1c*. Binding was determined to His-SUMO (grey), and to His-SUMO with fused ARR regions of CleB (pink), CleC (light blue), CleD (red) and CleE (dark blue), or with the ARR-CTR region of CleA (dark green). The respective $K_d$ values are indicated. (**d**) Sequence logo of the alignment generated by Jackhmmer when queried with the ARR region of CleD. Analysis of a representative pool of sequences (see Materials and Methods) identified the tandem motif [YF]XGP[DE][RK]R. The overall height of the stack indicates the sequence conservation at that position, while the height of symbols within the stack indicates the relative frequency of each amino acid at that position. Bits: Information content in bits.

DOI: https://doi.org/10.7554/eLife.28842.007

The following figure supplements are available for figure 3:

**Figure supplement 1.** Conservation and distribution of Rec-ARR proteins.
DOI: https://doi.org/10.7554/eLife.28842.008

**Figure supplement 2.** Conservation and distribution of Rec-ARR proteins.
DOI: https://doi.org/10.7554/eLife.28842.009

EcCheY-ARR revealed a $K_d$ of 150 nM for c-di-GMP and a 2:1 binding stoichiometry (*Figure 3—figure supplement 1a*).

Because full-length variants of the other Cle proteins were not soluble in vitro, we set out to analyze c-di-GMP binding to their isolated ARR peptides. For this purpose, ARR sequences of all Cle proteins were fused to the C-terminus of His-SUMO and used in MST experiments. As shown in *Figure 3c* we found that all ARRs but not SUMO itself were able to bind flr-cdG. The only exception was CleA, the isolated ARR of which failed to bind flr-cdG, but showed binding when purified together with its CTR. Thus, these experiments disclosed the short C-terminal ARR extensions of Cle proteins as high-affinity binding modules for c-di-GMP that retain their binding properties when grafted onto a heterologous carrier protein.

## The conserved ARR c-di-GMP binding motif defines a novel class of CheY proteins

To assess the natural occurrence of ARR-based c-di-GMP binding modules in proteins present in the databases, we employed iterative search approaches using Jackhmmer (*Finn et al., 2011*) and PSI-Blast (*Altschul et al., 1997*). Using the ARR sequence of CleD as query, we found homologous sequences in a total of 302 bacterial proteins including 294 from *alphaproteobacteria*, three from *Cyanobacteria* species and five from unclassified bacteria (*Figure 3—figure supplement 2a*). Most of these proteins (91%) are CheY homologs with a single Rec domain and a C-terminal ARR. Nine proteins harbor additional domains annotated as cNMP binding, hemerythrin, or adenylsucc_synt, and 5% of the hits do not contain any known domains. Significant hits (94, e-value <0.003) from the rp75 (UniProt Representative Sets (*UniProt Consortium, 2015*) were used to create a representative alignment of the ARR. This revealed a conserved motif [Y/F]XGP[D/E][R/K]R arranged in tandem as core feature of the ARR sequences flanked by a highly conserved Arg at the N- and an Asp at the C-terminus (*Figure 3d*).

Since Arg residues play key roles in coordinating c-di-GMP (*Chou and Galperin, 2016*) we analyzed the role of the conserved Arg residues of the CleD ARR. Indeed, the R156A mutant peptide showed strongly reduced binding to flr-cdG in MST experiments either when fused to CheY from *E. coli* ($K_d$471 µM) (*Figure 3b*) or when fused to SUMO (*Figure 3—figure supplement 1b*). Binding to the R156A mutant was completely abolished, while binding to the R169A mutant was reduced about 5-fold (*Figure 3—figure supplement 1b,c*). Although the ARR peptide appears to be sufficient for c-di-GMP binding, it is possible that conserved residues of the Rec domain also contribute to ligand interaction. We noticed that all Cle orthologs share a conserved Arg residue within helix α4 of the Rec domain (*Figure 1—figure supplement 1b*, *Figure 3—figure supplement 2b*). Mutating the corresponding residue R113 of CleD to Ala resulted in a 3-fold increase of the $K_d$ (221 nM) as compared to wild-type CleD (*Figure 3—figure supplement 1d*). Thus, the Rec domain may contribute to c-di-GMP binding in vivo (see below). Together, these data define a core motif for c-di-GMP binding in the Arg-rich peptide of the Cle protein family.

## Cle proteins interact with the flagellar basal body upon c-di-GMP binding

Given their close relationship with CheY we next asked if Cle proteins directly interact with the motor. We first analyzed the localization of CleA and CleD protein fusions to mGFPmut3 and eGFP, respectively. CleA was found at one cell pole in about 30% of the cells while 60% of the cells showed diffuse cytoplasmic localization (*Figure 4a,b*). CleA-mGFPmut3 foci were observed primarily at the pole opposite the stalk of late PD cells and in SW cells, arguing that CleA dynamically positions to the *C. crescentus* flagellated pole immediately before and after cell division. Similarly, CleD-eGFP was localized to the flagellated pole in roughly 30% of PD and swarmer cells (*Figure 4—figure supplement 1a*).

While polar localization of CleA and CleD was not dependent on their potential phosphoryl acceptor residues, mutants lacking the central Arg residues of the ARR peptide failed to localize to the pole. The more peripheral Arg167 was not required for CleA localization, while mutation of its counterpart R169 in CleD abrogated polar localization. Likewise, the conserved Arg positioned in the α4 helix of the Cle Rec domains (R111 in CleA; R113 in CleD) was strictly required for subcellular positioning of these proteins suggesting that residues of the core Rec domain might play a role in the activation of Cle proteins by c-di-GMP (*Figure 4b*, *Figure 4—figure supplement 1a*). This

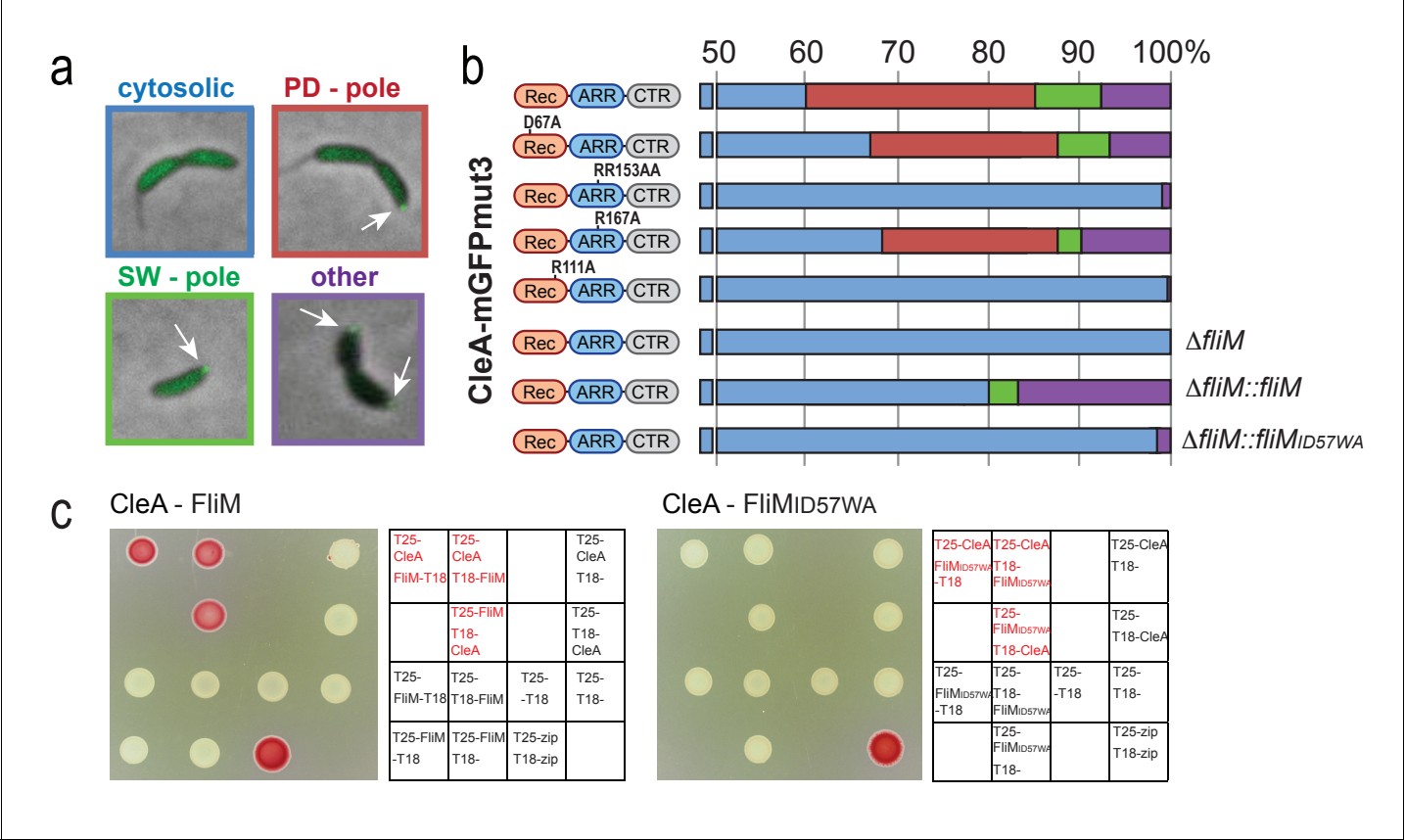

**Figure 4.** CleA localizes to the flagellated cell pole to interact with FliM. Examples of the subcellular localization of CleA-mGFPmut3 are shown in (**a**) and quantified as shown in (**b**). Variants of CleA fused to mGFPmut3 are indicated on the left with the three domains highlighted as in *Figures 2* and *3*. Individual amino acid substitutions are indicated. CleA fusion proteins were expressed in a Δ*cleA* deletion strain and additional genetic alterations as indicated on the right of each panel. Cells were classified according to the localization patterns shown in (**a**): delocalized cytosolic (blue); foci at flagellated pole of PD cells (red); polar foci in small SW cells (green); other localization patterns (purple), including PD cells with foci at both poles, PD cells with foci at stalked pole, cells with no GFP signal. Number of cells analyzed (top to bottom): 3452, 1450, 637, 2357, 2603, 795, 1148, 1347. (**c**) CleA interacts with the FliM flagellar switch protein. Binding of CleA to the N-terminus of FliM was determined by bacterial two-hybrid analysis. Fusion proteins between the T25 and T18 fragments of adenylate cyclase (*Karimova et al., 1998*) and the N-terminal peptide of FliM or FliM_ID57WA (aa 2–65), or full-length CleA were constructed and expressed in the *E. coli cya* mutant strain AB1770. Strains harboring combinations of FliM and CleA fusions were spotted on McConkey agar plates to score for interaction. Red color indicates reporter gene expression from a cAMP-dependent promoter. The spotting order is indicated in the grids on the right. A positive control with the adenylate cyclase fragments fused to the leucine zipper region of the yeast GCN4 protein (zip) is shown (lower right spots).

DOI: https://doi.org/10.7554/eLife.28842.010

The following figure supplements are available for figure 4:

**Figure supplement 1.** CleD localizes to the flagellated cell pole to interact with FliM.

DOI: https://doi.org/10.7554/eLife.28842.011

**Figure supplement 2.** The conserved CheY binding site of the *C. crescentus* FliM switch protein is required for motor reversal.

DOI: https://doi.org/10.7554/eLife.28842.012

indicated that an intact c-di-GMP binding site in the ARR peptide is required for the localization of CleA and CleD to the flagellated cell pole.

Because CleA and CleD localize to the cell pole, we next asked if these proteins, like their CheY homologs, bind to the flagellar switch protein FliM. In *E. coli* CheY interacts with a widely conserved CheY binding motif (LSQxEIDxLL) located at the N-terminus of FliM (*Bren and Eisenbach, 1998*; *Lee et al., 2001*). This interaction surface is conserved in *C. crescentus* (*Figure 4—figure supplement 2a,b*). When the highly conserved Ile/Asp motif at position 57/58 of FliM was replaced with a bulky Trp and an uncharged Ala residue, respectively, overall motility and swimming speed was maintained but cells failed to reverse their swimming direction (*Figure 4—figure supplement 2c*).

This is in line with the expected failure of CheY to interact with the FliM motor switch to cause motor reversals. Importantly, CleA and CleD also failed to localize in the *fliM_ID57WA* mutant (*Figure 4b*; *Figure 4—figure supplement 1a*). Moreover, bacterial two-hybrid assays (*Karimova et al., 1998*) showed strong interaction of CleA with the N-terminal peptide of FliM (amino acid 2 to 65) but not with the FliMID57WA mutant peptide (*Figure 4c*). Although bacterial two-hybrid experiments failed to show interaction between CleD and FliM, co-immunoprecipitation (co-IP) experiments showed co-purification of CleD-Flag with FliM (*Figure 4—figure supplement 1b*). Likewise, NMR experiments confirmed CleD binding to synthetic FliM peptides corresponding to residues 47 to 62 of FliM. A $^1$H-$^{15}$N HSQC spectrum of $^{15}$N-CleD was recorded alone and in the presence of c-di-GMP and the FliM wild-type peptide in two-fold excess. The latter resulted in shift changes of CleD of approximately 10 resonances (*Figure 4—figure supplement 1c*). In contrast, neither the addition of the Fli-MID57WA mutant peptide with c-di-GMP nor the FliM wild-type peptide without c-di-GMP gave an observable chemical shift.

Thus, CleA and CleD bind to the FliM switch protein in a way that is similar or identical to CheY docking to the flagellar motor. Based on this and on the data shown above, we conclude that this interaction is regulated by c-di-GMP binding to CleA and CleD.

## CleA interferes with the *C. crescentus* chemotaxis response

The findings that the Cle proteins phylogenetically cluster with CheY chemotaxis proteins, that they interact with FliM, and that one of the *cle* genes, *cleA*, is part of a large chemotaxis operon in *C. crescentus* prompted us to analyze a potential role of Cle proteins in motility behavior. Strains with single *cle* deletions showed increased spreading on semisolid agar plates (*Figure 5a*). Strikingly, spreading increased almost four-fold in a mutant lacking all five Cle proteins (*Figure 5a*). This finding was surprising since *cheY* mutants normally exhibit reduced performance on semisolid agar plates due to aberrant chemotaxis behavior (*Wolfe and Berg, 1989*). Indeed, a mutant lacking CheYII, the prototypical *C. crescentus* CheY encoded in the *che1* cluster (*Ely et al., 1986*) (*Figure 1b*) showed very poor spreading on semisolid agar plates (*Figure 5a*). To investigate this, we first set out to analyze swimming behavior of *C. crescentus cleA* mutant in a homogenous environment. In order to analyze swimming behavior without surface interference, we assayed a Δ*pilA* mutant background that lacks adherent pili. Cells were placed into a quasi-2D environment between glass slide and cover slip and their swimming trajectories were recorded at 16 frames per second for 30 s. As baseline we compared the swimming behavior of a *C. crescentus* Δ*pilA* mutant with mutants lacking all chemoreceptor-dependent methyltransferases (Δ*cheR*1-3) or methylesterases (Δ*cheB*1-2) (*Briegel et al., 2011*). In accordance with their role in chemotaxis (*Sourjik and Wingreen, 2012*) the Δ*cheR* mutant showed strongly reduced reversal rates (*Briegel et al., 2011*), while the Δ*cheB* mutant had increased reversal rates (*Figure 5b*). The Δ*cleA* mutant and a strain lacking all five Cle proteins (Δ*cleA-E)* showed strongly increased reversal frequencies and reduced swimming speeds (*Figure 5b*). Likewise, tethered cells of the *cleA* deletion strain had strongly increased reversal rates as compared to the *cleA*$^+$ control strain (*Video 1*, *Video 2*). In contrast, strains lacking CleD or lacking all Cle proteins except CleA showed normal reversal frequencies (*Figure 5—figure supplement 1a,b*).

These observations suggested that CleA is involved in *C. crescentus* chemotaxis and that its role is to prevent flagellar reversals and to promote smooth swimming. This was unexpected since canonical CheYs generally induce motor reversals (*Smith et al., 1988*; *Parkinson and Houts, 1982*). CleA is encoded in chemotaxis cluster *che1* together with two CheY paralogs, CheYI and CheYII (*Figure 1b*). Consistent with a major role in chemotaxis, a mutant lacking CheYII was unable to spread on semisolid agar plates (*Skerker et al., 2005*) (*Figure 5a*) and showed significantly reduced reversal rates (*Figure 5c*). In contrast, a *cheYI* mutant exhibited a hyper-reversal phenotype, akin to the *cleA* mutant, but with no significant drop in swimming speed (*Figure 5c*). Importantly, double mutants lacking CheYII and either CheYI or CleA, performed poorly on motility plates (*Figure 5a*) and showed reduced reversal rates, similar to the *cheYII* mutant (*Figure 5c*, *Figure 5—figure supplement 1c*). Together, these results suggested that CleA, CheYI and CheYII tune *C. crescentus* motor reversals. The data also argued that CheYII adopts a dominant, canonical CheY-like role to induce motor reversals, while CheYI and CleA seem to antagonize CheYII activity to promote smooth swimming.

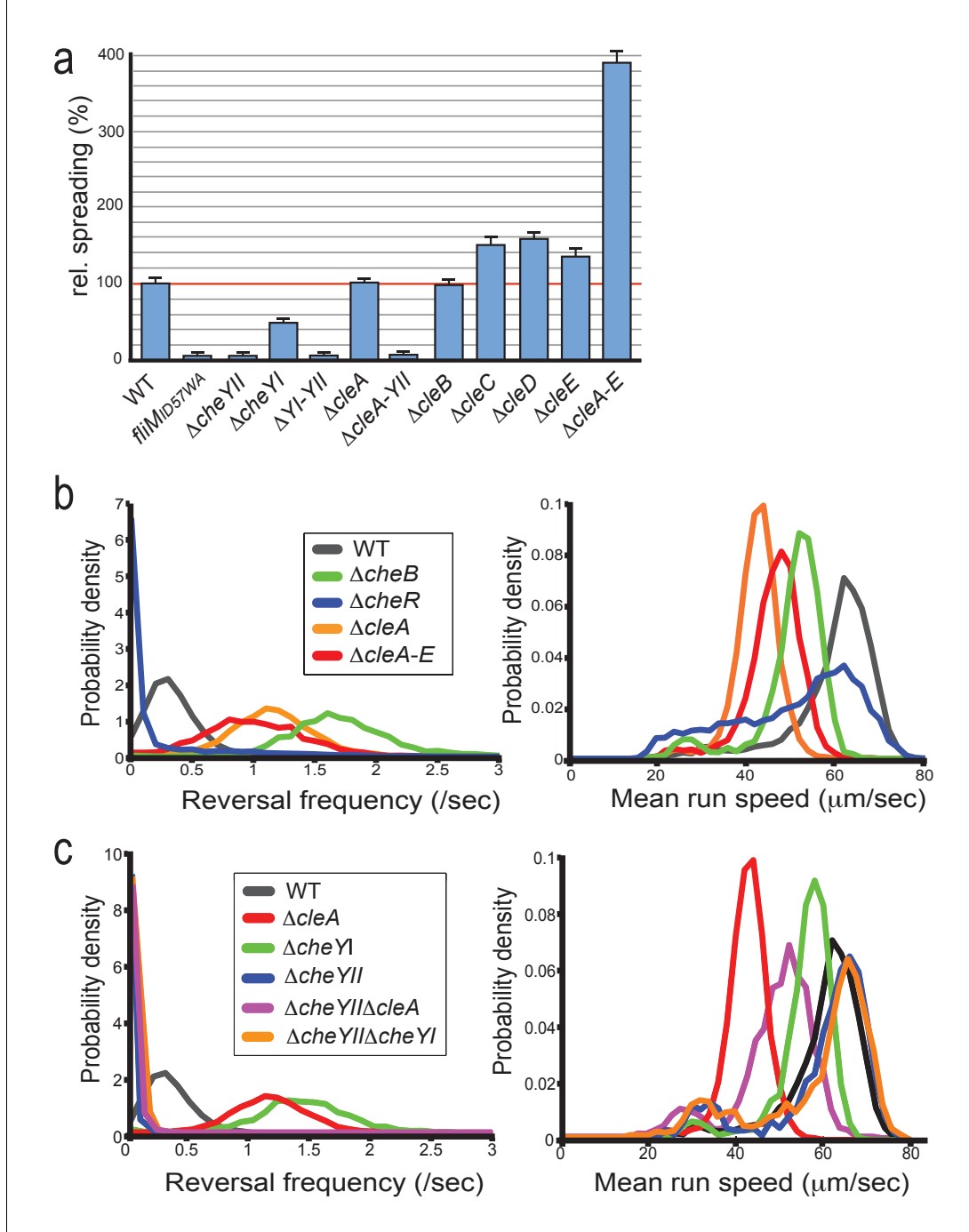

**Figure 5.** CleA tunes the flagellar motor by interfering with the chemotaxis response. (**a**) Cle proteins impede spreading on semisolid agar. Relative spreading areas of *C. crescentus* wild type and *fliM*, *cheY* and *cle* mutants are indicated. Strains were incubated at 30°C and motility was scored after 72 hr as the overall area covered by individual strains. Strains containing single or multiple deletions are indicated. (**b**) Mutants lacking CleA show a hyper-reversal phenotype. Directional reversal frequencies and mean run speed were measured for individual swimming cells of *C. crescentus* wild type and mutants lacking CheB, CheR, CleA, or all five Cle proteins. All strains analyzed harbored an additional deletion in the *pilA* gene to avoid pili-mediated motility variations. Cells were located in a pseudo-2D environment and their swimming trajectories were recorded at 16 frames per second. The fraction of cells with a given reversal frequency (left) or mean run speed (right) is indicated. (**c**) CleA promotes smooth swimming by antagonizing the major *C. crescentus* CheY, CheYII. Analysis of reversal frequencies and mean speed of the strains indicated was as in (**b**). It should be noted that the reversal frequency profiles of strains ΔcheYII, DcheYIIΔcleA and ΔcheYIIΔcheYI overlay. Experiments in (**b**) and (**c**) include the analysis of >850 cells for each strain.

DOI: https://doi.org/10.7554/eLife.28842.013

*Figure 5 continued on next page*

*Figure 5 continued*

The following figure supplement is available for figure 5:

**Figure supplement 1.** Reversal frequencies and speed of *C. crescentus* mutants lacking specific CheY and Cle components.
DOI: https://doi.org/10.7554/eLife.28842.014

From these data, we concluded that CheYII is the major CheY protein involved in *C. crescentus* chemotaxis and that CleA, together with CheYI, modulates motor reversals, possibly by competing with CheYII for the same motor binding sites. It is unclear why the *cleA* mutant, despite of its strong influence on motor reversals, showed normal performance on semisolid agar plates (*Figure 5a*). Possibly, the strongly increased reversal rates observed for these mutants in a homogenous environment may be adjusted in a chemical gradient by the activity of CheYII or additional functional CheY homologs.

## Cle proteins promote rapid surface attachment

While the above studies proposed a role for CleA in motility control, none of the other Cle proteins appeared to modulate motor reversal or swimming speed under the conditions tested (*Figure 5— figure supplement 1a,b*). At the same time, deleting all *cle* genes strongly boosted spreading in low percentage agar (*Figure 5a*). We thus hypothesized that some of the Cle proteins mediate physiological changes sparked by (agar) surface contact. This is based on the premise that cells moving through semisolid agar will be challenged by physical cues and may respond by reducing their motility and boosting their propensity to attach. To test if the observed attachment defect of *cle* mutants resulted from a reduced capability to sense and respond to surface interactions, we used microfluidic-based assays that allow to directly score surface attachment of SW cells (*Persat et al., 2014*). When dividing cells are cultured under strong medium flow, the newborn swarmer progeny rapidly deploy an adhesive holdfast upon surface contact and thus are able to adhere before being washed out (*Hoffman et al., 2015*; *Hug et al., 2017*) (*Figure 6a,b*). In agreement with earlier findings, an active flagellar motor is strictly required for rapid holdfast formation under these conditions (*Figure 6a*) (*Li et al., 2012*; *Hoffman et al., 2015*; *Hug et al., 2017*). A mutant lacking all Cle proteins (Δ*cleA-E*) showed severely impaired attachment. When *cleC*, *cleD* or *cleE* were expressed in trans surface colonization of the Δ*cleA-E* strain was restored (*Figure 6a*). Importantly, restoring surface attachment under these conditions required a motor with intact FliM docking sites for Cle proteins (*Figure 6a*).

Finally, we made use of microfluidic devices with small, quasi-2D side chambers (0.75 μm in height) to enhance surface encounter of newborn swarmer cells (*Hug et al., 2017*). This assay allows following SW progeny microscopically from the moment of birth at division to the onset of holdfast production. The tight geometry of such micro-chambers offers cells constant surface interaction opportunities without medium flow (*Figure 6c*). Hence, wild-type cells generally deployed a visible holdfast within 1–2 min after separating from their mothers, while cells lacking a functional motor ($motB_{D33N}$) (*Zhou et al., 1998*; *Kojima and Blair, 2001*) showed a significant delay in holdfast formation (*Figure 6d*) (*Hug et al., 2017*). Mutants devoid of the Cle proteins (Δ*cleA-E*) or cells lacking a functional Cle docking site on FliM ($fliM_{ID57WA}$) showed delayed holdfast formation. Importantly, epistasis analysis revealed that the motor is dominant over *cle* deletions, arguing that Cle proteins are not essential for mechanosensation per se, but rather improve the efficiency of the surface response (*Figure 6d*).

Together, these data suggested that a subset of the Cle proteins interacts with the flagellar motor to promote rapid surface attachment of *C. crescentus* swarmer cells.

## Cle activation requires c-di-GMP binding but not phosphorylation

The data in *Figure 4* indicated that c-di-GMP binding but not phosphorylation is important for polar localization and activation of CleA and

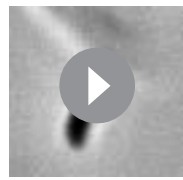

**Video 1.** NA1000 Δ*pilA* tethered on cover slips.
DOI: https://doi.org/10.7554/eLife.28842.015

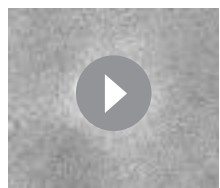

**Video 2.** Movie of NA1000 ΔpilA ΔcleA deletion mutant tethered on cover slips.
DOI: https://doi.org/10.7554/eLife.28842.016

CleD. To scrutinize the activation mechanism of CleA more closely, we compared the behavior of mutants lacking the conserved phospho-acceptor Asp residue with mutants unable to bind c-di-GMP. Different *cleA* alleles were expressed on a plasmid in a Δ*cleA* mutant (*Figure 7a*). Expression of wild-type *cleA* and the *cleA_D67A* allele restored running speed and low reversal frequencies of a Δ*cleA* mutant under free-swimming condition in a homogeneous environment (*Figure 7b*). In contrast, the *cleA_RR153AA* allele was non-functional indicating that c-di-GMP binding is important for CleA function. Likewise, R111 seems to contribute partially to CleA function (*Figure 7c*, *Figure 7—figure supplement 1a*).

To assay CleD function, we made use of the observations that cell spreading in semisolid agar is strongly inhibited at elevated c-di-GMP concentrations, i.e. in a mutant lacking the phosphodiesterase PdeA (*Abel et al., 2011*), and that a Δ*cleD* deletion partially rescued this defect (*Figure 7—figure supplement 1b*). A plasmid-born copy of *cleD* reinstated the spreading block in a *cleD pdeA* double mutant (*Figure 7—figure supplement 1b*). Likewise, expression of the *cleD_E70A* allele reduced motility, demonstrating that this CleD variant is fully functional. In contrast, an intact ARR and the highly conserved Arg in helix α4 were strictly required for CleD function (*Figure 7—figure supplement 1b*).

In summary, these results demonstrated that CleA and CleD are activated by c-di-GMP binding, while conserved residues of the phosphorylation-mediated receiver domain switch (*Volkman et al., 2001*) are dispensable for Cle activation. We speculate that Cle proteins have lost their association with CheA-mediated phosphorylation and instead have adopted c-di-GMP-mediated control by the acquisition of the C-terminal ARR peptide.

## Discussion

We used a chemical proteomics approach to identify a novel class of CheY-like proteins that are activated by binding of the second messenger c-di-GMP. Two of these proteins, CleA and CleD, were shown to interact with the flagellar motor via the canonical CheY binding site on the flagellar switch. Given the strong conservation and clustering of these proteins with other *C. crescentus* CheY proteins, we propose that all five Cle proteins interact with the polar flagellar motor, possibly in a cooperative or competitive manner with conventional CheY proteins (*Figure 1c*). These findings raise several questions. How does c-di-GMP activate the CheY-like receiver domain to interact with the motor switch? Second, what is the role of individual Cle proteins in modulating motor function and how do they integrate with the canonical CheY-driven chemotaxis response? And finally, how can individual Cle components elicit a specific response at the flagellar motor?

### Mechanism of Cle activation by c-di-GMP

The c-di-GMP binding site of Cle proteins was confined to a 28-amino acid extension of their conserved CheY-like receiver domains. This sequence contains a tandem repeat of the motif [Y/F]XGPX[R/K]R (*Figure 3d*). In many c-di-GMP effectors Arg residues are prominently involved in the coordination of the four guanine rings of an intercalated ligand dimer (*Chou and Galperin, 2016*). The tandem arrangement of the Arg-containing motifs together with the binding affinity and stoichiometry determined for CleD suggested that all Cle proteins bind a dimer of c-di-GMP and that conserved Arg residues play a central role in this process. Structural analysis will eventually reveal how exactly this novel binding motif intercalates c-di-GMP.

Although ARR peptides comprise high affinity binding sites for c-di-GMP that can be grafted onto other proteins, our data identified at least one conserved Arg residue positioned in helix α4 of the receiver domain that contributes to c-di-GMP binding and activation of the Cle proteins. Activation of canonical CheY receiver domains by phosphorylation results in subtle conformational changes of the α4-β5-α5 surface, which promote their interaction with the FliM switch (*Lee et al., 2001*; *Gao and Stock, 2010*). In analogy, it is plausible that the rearrangement of the ARR peptide via a

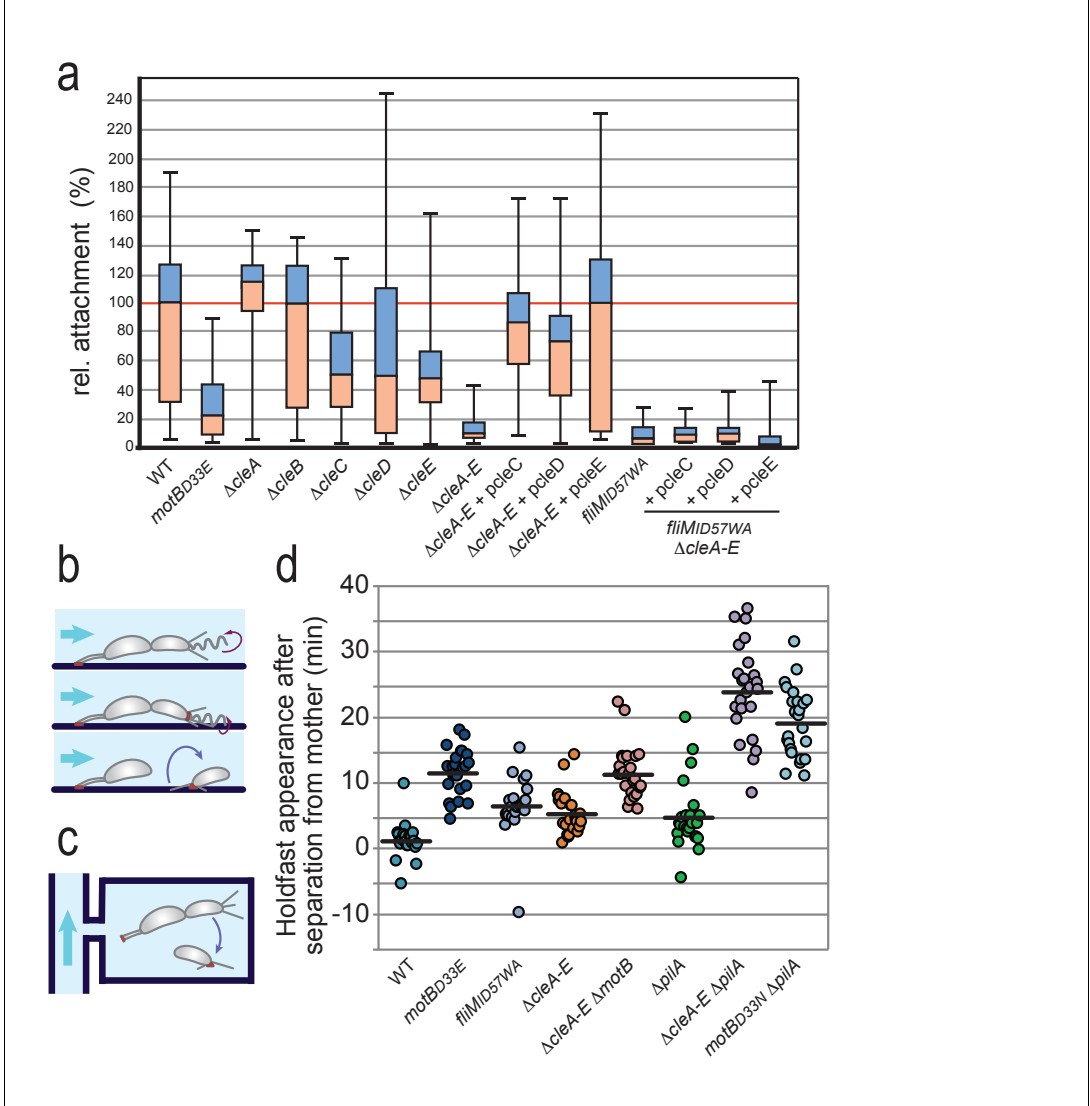

**Figure 6.** Role of Cle proteins in *C. crescentus* surface attachment. (**a**) Cle proteins promote rapid surface attachment under flow. *C. crescentus* wild-type and mutant strains were assayed as indicated in (**b**). The efficiency of rapid surface attachment of newborn SW progeny was determined as the relative area covered by microcolonies emerging from a single mother cell after 15 hr incubation. Box plots mark the median (horizontal black lines), the lower and upper quartiles (red and blue boxes) and the extreme values (whiskers). P values obtained with a 2-tailed T-test were < 0.05 for Δ*cleC* and Δ*cleE*, whereas the Δ*cleA*,Δ*cleB* and Δ*cleD* mutants were statistically not significant. (**b**) Experimental setup for microfluidics experiments shown in (**a**). The direction of medium flow in the microfluidic channels is indicated by blue arrows. The position of holdfast (red), pili and flagella are indicated. Newborn wild type SW cells are able to sense surface exposure and rapidly synthesize a holdfast before cell separation and remain attached downstream of their mother cells (*Hug et al., 2017*). (**c**) Experimental setup for experiments shown in (**d**). Dividing *C. crescentus* cells were trapped in narrow quasi-2D chambers that offer immediate surface contact in the absence of flow. Holdfast was visualized using Oregon-green labeled wheat germ agglutinin lectin. Time-lapse imaging at four frames per minute allowed the determination of the time elapsed from separation of individual SW cells and the first detection of their newly secreted holdfast (*Hug et al., 2017*). (**d**) Timing of holdfast appearance after separation from mother cells. Dots represent individual cells of *C. crescentus* wild type and mutants as indicated. Horizontal black lines mark the average values. Values for Δ*cleA-E* and *fliMID57WA* mutants were significantly different from both WT and *motBD33N* with P values with a 2-tailed T-test < 0.01.

DOI: https://doi.org/10.7554/eLife.28842.017

specific c-di-GMP docking site in helix α4 remodels the α4-β5-α5 surface of Cle proteins to promote its interaction with the motor switch. Such a mechanism would abrogate the need for phosphoryla-tion-mediated structural changes, explaining why Cle proteins no longer depend on conserved residues of the phospho-switch (*Gao and Stock, 2010*). It was proposed recently that second messenger-based control may have expanded rapidly during bacterial evolution through the

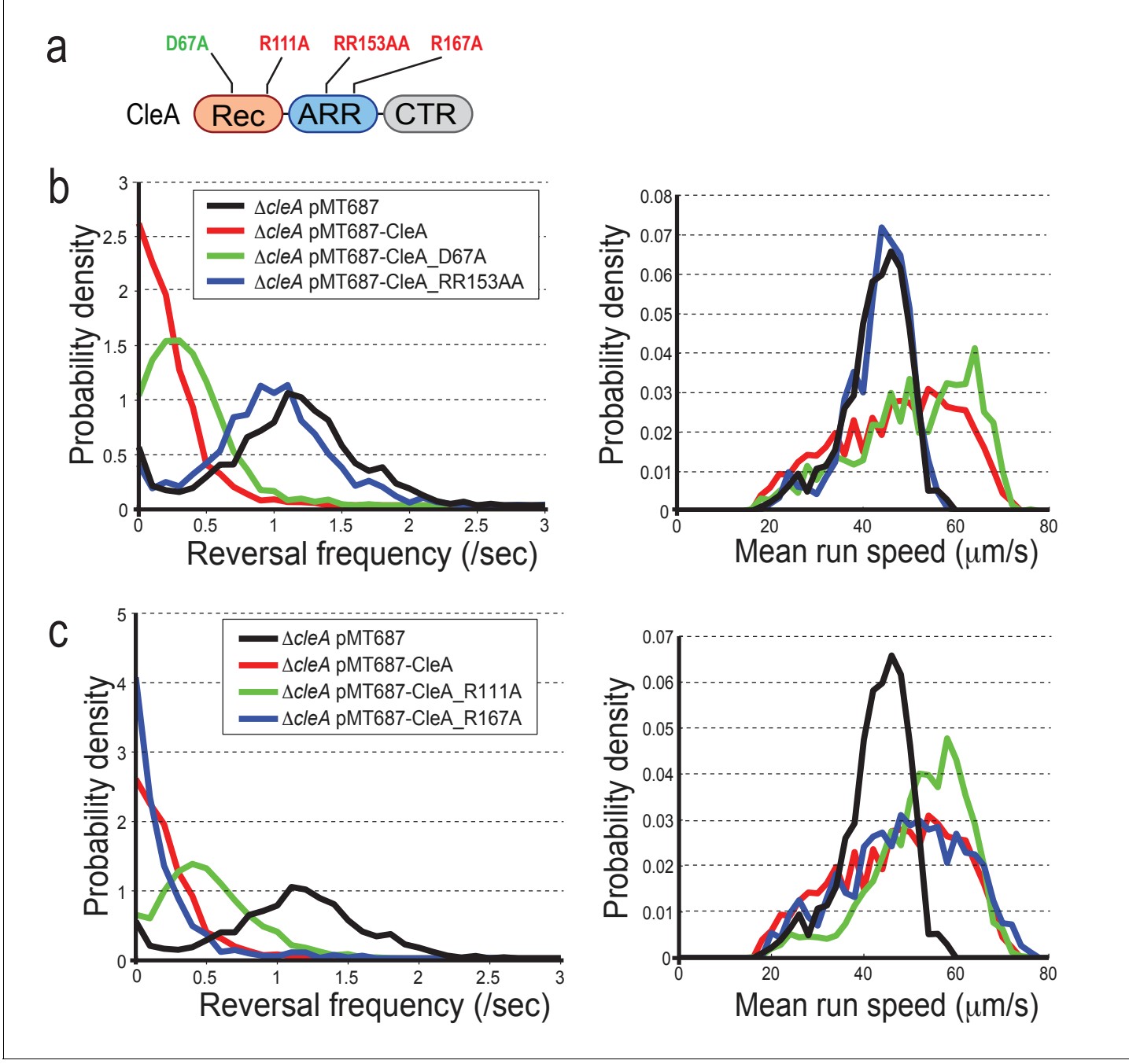

**Figure 7.** CleA is activated by c-di-GMP binding but not phosphorylation. (**a**) Schematic of the domain structure of CleA with mutations that interfere with potential phosphorylation control (green) or c-di-GMP binding (red). The domains are labeled as in *Figures 2*, *3* and *4*. (**b-c**) CleA is activated by c-di-GMP binding. Single cell analysis of reversal frequencies and mean speed of *C. crescentus* wild type and Δ*cleA* mutants harboring plasmid driven *cleA* alleles as indicated in (**a**). *cleA* alleles were expressed from a xylose-dependent promoter and were induced with 0.1% xylose for 3 hr before imaging. Experiments in (**b**) and (**c**) include the analysis of >850 cells for each strain.

DOI: https://doi.org/10.7554/eLife.28842.018

The following figure supplement is available for figure 7:

**Figure supplement 1.** CleA and CleD are activated by c-di-GMP binding but not phosphorylation.

DOI: https://doi.org/10.7554/eLife.28842.019

recruitment of additional cellular components into existing c-di-GMP networks (*Jenal et al., 2017*). The ease with which diverse cellular processes can be interconnected may have predisposed c-di-GMP for the coordination of complex behavioral transitions. Accordingly, regulatory proteins like CheY may have been captured during evolution by c-di-GMP to allow the second messenger to adopt flagellar motor control.

## Role of cle proteins in *Caulobacter* cell behavior

The discovery of Cle proteins and their interaction with the *C. crescentus* motor adds to the growing evidence that c-di-GMP can modulate bacterial chemotaxis (*Paul et al., 2010*; *Fang and Gomelsky, 2010*; *Russell et al., 2013*; *Xu et al., 2016*) and motor performance (*Boehm et al., 2010*; *Chen et al., 2012*; *Baker et al., 2016*) (*Figure 1c*). But how does this multi-component system interfere with the canonical CheY-mediated flagellar switch and what is it good for? Phenotypic analysis revealed that individual Cle proteins have distinct functions in motility and surface adaptation, suggesting that they represent discrete signaling pathways affecting motor activity. Individual Cle proteins may differ in their interaction with the motor or with conventional CheYs. The specific functions of Cle proteins may involve their distinct C-terminal extensions (*Figure 1—figure supplement 1b*), which could help sequester partner proteins with specific functionality to the motor. All five *cle* genes are expressed in PD cells, coincident with the transcription of flagellar and chemotaxis genes (*Schrader et al., 2014*; *Zhou et al., 2015*). The concentration of c-di-GMP fluctuates during the cell cycle with c-di-GMP being low in SW cells and reaching peak levels during the SW-to-ST cell transition (*Abel et al., 2013*; *Christen et al., 2010*). Recent evidence suggested that c-di-GMP is not evenly distributed in dividing cells with a spatial trough being established at the pole where the flagellum is assembled (*Lori et al., 2015*). Thus, the c-di-GMP distribution is highly dynamic making it difficult to predict the interaction of Cle proteins with the flagellar motor. Specific Cle proteins may be active only during very short time windows when cells enter or exit motility and when levels of c-di-GMP rapidly change. Accordingly, one or several of these proteins may coordinate flagellar rotation with flagellar assembly or motor dismantling (*Abel et al., 2013*; *Aldridge and Jenal, 1999*; *Aldridge et al., 2003*), a potentially short-lived state difficult to capture microscopically. In line with this, mutants unable to effectively increase c-di-GMP levels during the SW-to-ST transition fail to eject the rotary motor and retain motility for an extended period (*Aldridge and Jenal, 1999*; *Aldridge et al., 2003*).

CleA regulates reversal frequencies of swimming cells suggesting that it interferes with chemotaxis control. The *cleA* gene is part of the *che1* chemotaxis cluster, which also contains two *cheY* homologs, *cheYI* and *cheYII*. Because both CheYI and CheYII lack a c-di-GMP binding site and have conserved phospho-switch residues, they are likely regulated through chemoreceptor-mediated phosphorylation. Our experiments identified CheYII as a prototypical CheY-like component promoting motor reversals. In contrast, both CheYI and CleA favor smooth swimming. Epistasis experiments strongly suggested that CheYI and CleA antagonize CheYII activity possibly by competing for the same binding sites on the flagellar motor switch (*Figure 1c*). While CheYII is essential for directional motility in gradients of nutrients, the role of CheYI and CleA is less clear. Based on our findings we propose that both proteins, in response to yet unknown cues, promote smooth swimming by competing with CheYII for motor binding sites.

CleA also influences motor speed. This is reminiscent of motor control in *E. coli*, where YcgR upon c-di-GMP binding influences both motor reversal and speed (*Boehm et al., 2010*; *Paul et al., 2010*; *Fang and Gomelsky, 2010*). How exactly this is accomplished and how these two proteins are controlled, remains to be investigated. We anticipate that c-di-GMP levels fluctuate in SW cells to dynamically activate CleA without reaching threshold concentrations required to fully induce the motile-sessile switch and to drive bacteria into permanent attachment, a scenario that was also proposed for other bacteria (*Russell et al., 2013*; *Kulasekara et al., 2013*). Such a mechanism could for instance promote near-surface swimming, a phenomenon that can trap bacteria transiently near surfaces and facilitate surface 'scanning' (*Misselwitz et al., 2012*; *Molaei et al., 2014*; *Jones et al., 2015*). In line with this, it was proposed recently that c-di-GMP can counteract CheY ~P in enteric bacteria, thereby promoting smooth swimming and helping bacteria to interact with surfaces (*Fang and Gomelsky, 2010*; *Girgis et al., 2007*).

But the role of the flagellar motor upon surface interaction is not merely a passive one. Several studies have inferred that flagella can serve as a mechanosensitive devices to assist surface

adaptation (*Harshey and Partridge, 2015*; *Belas, 2014*). Motile *C. crescentus* cells use their flagellum as a mechanosensitive device to instigate rapid production of the exopolysaccharide-based holdfast glue (*Li et al., 2012*; *Hoffman et al., 2015*; *Hug et al., 2017*). We have shown recently that a specific diguanylate cyclase, DgcB, is activated upon surface-induced motor interference. The c-di-GMP upshift in turn allosterically activates the holdfast synthesis machinery (*Hug et al., 2017*). Here we show that a subset of the Cle proteins is involved in mediating this rapid *C. crescentus* surface response. Mutants lacking one or several of these Cle proteins performed much better when spreading through semisolid agar. This is probably not due to interference with swimming or chemotaxis, as with the exception of *cleA* all mutants showed normal reversal rates and running velocities. Rather, we propose that the improved motility observed for some of these mutants was caused by their inability to respond to agar surface in motility plates. In support of this, mutants lacking CleC and CleE, but not CleA, showed increased response times in synthesizing adhesive holdfast when challenged with surface. Accordingly, their ability to rapidly colonize surfaces under medium flow was strongly reduced. Together with the observation that a mutant lacking the motor binding sites for Cle proteins also underperformed in rapid surface attachment, these data proposed that some Cle proteins may be part of a regulatory loop controlling flagellar activity or conformation in response to a transient surface-mediated upshift of c-di-GMP. Cle proteins may be activated upon surface contact by an initial DgcB-mediated boost of c-di-GMP levels leading to altered motor behavior promoting surface colonization. Moreover, activated Cle proteins may form a positive feedback loop by interacting with the mechanosensing device itself to reinforce the c-di-GMP upshift and to robustly and rapidly activate holdfast biogenesis.

Altogether, our studies discover an additional layer of a highly complex interaction network between cellular signaling components and the flagellar motor switch. These interactions not only interfere with chemotaxis but also control additional motor behavior to optimize the interaction of motile cells with surfaces. Since c-di-GMP is a key regulatory component coordinating the behavior of bacterial cells on surfaces, we propose that during evolution this molecule has hijacked CheY-mediated motor control to appropriately tune this rotary device to optimally approach and colonize biotic or abiotic surfaces. Together, the Cle and CheY proteins may integrate information from mechanical and chemical stimuli to direct motor behavior.

# Materials and methods

## Strains, plasmids, and growth conditions

The bacterial strains, plasmids and oligos used in this work are summarized in *Supplementary file 1A, B and C* respectively. *E. coli* was grown in Luria Broth (LB) media at 37°C and *C. crescentus* was grown in rich medium (peptone yeast extract; PYE) at 30°C (*Ely, 1991*). Marker-less deletions in *C. crescentus* were generated using the standard two-step recombination sucrose counter-selection procedure based on pNPTS138-derivatives. *E. coli* S17.1 was used to transfer plasmids by conjugation into *C. crescentus* strains (*Ely, 1991*). Motility behavior of *C. crescentus* was scored on semisolid PYE agar plates (0.3%). For synchronization experiments cells were grown at 30°C in minimal medium containing 0.2% glucose (*Ely, 1991*) and swarmer cells were isolated after Ludox gradient centrifugation (*Jenal and Shapiro, 1996*). For induction of plasmids with a vanillate inducible promotor a concentration of 1 mM vanillate, and with a xylose inducible promoter 0.1% xylose was added to the media. If not stated otherwise, exponentially growing cells were used for all experiments. Antibiotics were used at the following concentrations: Kanamycin 50 µg/ml (*E. coli*) and 5 µg/ml (*C. crescentus*); Chloramphenicol 30 µg/ml (*E. coli*) and 2 µg/ml (*C. crescentus*); Tetracycline 12.5 µg/ml (*E. coli*) and 2.5 µg/ml (*C. crescentus*). The bacterial strains, plasmids and oligos used in this study are summarized in *Supplementary file 1A-C*.

## Fluorescence microscopy

For microscope imaging, exponentially growing cells were spotted on agarose pads (Sigma, 1% in water). A Delta Vision Core microscope (GE Healthcare) equipped with a UPlan FL N 100x Oil objective (Olympus, Japan) and a pco.edge sCMOS camera was used to take phase contrast and fluorescence images at 30°C. For GFP fluorescence, GFP filter sets (Ex 461–489 nm, Em 525–50 nm) were

used. The exposure time was set to 1 s for eGFP fusions and to 500 ms for mGFPmut3 constructs. Images were processed with softWoRx 6.0 (GE Healthcare) and ImageJ software.

The strains expressing CleD-eGFP or derivatives from a plasmid with a vanillate inducible promoter and CleA-mGFPmut3 or derivatives from a plasmid with xylose inducible promoter were compared in their ability to localize to the cell poles by fluorescence microscopy. The cells were counted and classified according to the presence or absence of polar foci. All strains were grown to an OD of 0.3 in PYE containing the appropriate antibiotics and then induced with 1 mM vanillate or 0.1% xylose for 3 hr before microscopic analysis.

## Protein expression and purification

Proteins were expressed in *E. coli* BL21 (DE3) (Stratagen) by adding 0.5 mM IPTG at an $OD_{600}$ of 0.5 and incubation for 6 hr at 37°C. Cells were collected by centrifugation, resuspended in lysis buffer (see below) and disrupted by a French pressure cell press (Thermo, Elcetron corporation). Lysates were centrifuged for 1 hr at 40000 rpm and proteins were purified using affinity chromatography.

For purification of StepII-tagged protein Strep-Tactin Superflow Plus (Quiagen) was used and batch purification was carried out according to the manufactures protocol using NP buffer for lysis and washing and NP containing 2.5 mM Desthiobiotin for elution. All His-tagged proteins were purified using a 1 mL HisTrap HP column (GE Healthcare) connected to a ÄKTA chromatography system. For lysis, 50 mM Tris pH8, 500 mM NaCl, 30 mM Imidazole, 5 mM $MgCl_2$ and 1 mM DTT, pH eight was used and proteins were eluted with increased Imidazole concentration up to 500 mM. Tagged proteins were further purified by size exclusion chromatography on a HiLoad Superdex S200 16/60 column (GE Healthcare) equilibrated with 50 mM Tris-HCl pH 8.0, 100 mM NaCl, 5 mM $MgCl_2$ and 1 mM DTT. PA4608 was purified as described recently (*Habazettl et al., 2011*).

## Capture compound mass spectrometry (CCMS)

CCMS was carried out using the c-di-GMP specific capture as described recently (*Nesper et al., 2012*; *Laventie et al., 2015*).

## UV cross-linking

Synthesis of radiolabeled c-di-GMP using YdeH and UV light-induced cross-linking experiments in conical 96-well plates (Greiner Bio-One) were performed as described recently (*Steiner et al., 2013*; *Christen et al., 2005*). Briefly, 2 µM protein was incubated for 10 min at room temperature with c-[$^{33}$P]-di-GMP in a total volume of 20 µl using PBS as buffer. As a control BSA was included. For competition experiments unlabeled nucleotides were incubated with the protein for 15 min prior the addition of c-[$^{33}$P]-di-GMP. The 96-well plates were then UV-irradiated at 254 nm for 20 min using a Bio-Link crosslinker (Vilber Lourmat, France). After addition of 5 µl loading dye and boiling for 5 min the samples were subjected to SDS-PAGE. Proteins were analyzed by Coomassie staining and autoradiography. Band intensities were quantified using the ImageJ64 software and the data were fitted using GraphPad Prism version 5.04 for Windows (GraphPad Software, San Diego California USA). The $K_d$ was calculated by non-linear regression using the 'One site – Total binding' equation.

## Immunodetection

To generate antibodies against CleD the purified His-protein was applied to polyacrylamide gel electrophoresis. After elution from the SDS gel the denatured protein was injected into rabbits for polyclonal antibody production (Josman, LLC, Californien, USA). The serum was adsorbed against a whole cell lysate of the *cleD* deletion mutant. For immunoblots anti-CleD antiserum was diluted 1:1000. Other antibodies were used in the following dilutions: anti-CtrA 1:10,000, anti-FliF 1:10,000, anti-CcrM 1:10,000, anti-DgcB 1:10,000, anti-FliM, anti-GFP (Invitrogen) 1:800, anti-Flag (Sigma) 1:10,000, HPR-conjugated swine α-rabbit antibodies (DAKO) 1:10000, or HPR-conjugated rabbit α-mouse antibodies (DAKO) 1:10,000. After incubation with ECL chemiluminescent substrate (Perkin Elmer, USA), X-ray films (Fujifilm Corporation) were used to detect luminescence.

## Co-IP using flag-tagged proteins

Cells were grown in 200 ml PYE to an $OD_{660}$ of 0.3. Cells were pelleted and lysed in 2 ml Bugbuster (Novagen). After centrifugation for 10 min at 16,000 g the supernatant was incubated O/N at 4°C

with 15 µl washed M2 coupled agarose beads (Sigma). The beads were washed 8 x with 200 µl Bug-buster and then resuspended in 30 µl SDS loading dye.

## Isothermal titration calorimetry (ITC)

ITC experiments were performed on a VP-ITC MicroCalorimeter (MicroCal, Northampton, MA, USA). Protein samples, c-di-GMP and buffer (50 mM Tris-HCl, pH 8.0, 100 mM NaCl, 5 mM MgCl$_2$, 1 mM DTT) were degassed for 15 min prior to filling into sample cell and syringe. The sample cell contained 7.4 µM of His-CleD-GFP and the syringe contained 69.7 µM of cdG. The concentrations were determined in a spectrophotometer using a cuvette with 1 cm path length. All experiments were carried out at 10 °C with 30 injections (10 µl each) with a 250 s interval. Stir rate was 300 rpm. The data were evaluated using ITC Data Analysis in ORIGIN (MicroCal) provided by the manufacturer. The area under each peak was integrated and fitted using the built-in one-site model by non-linear least squares regression analysis.

## Microscale thermophoresis

MST experiments were carried out in a Monolith NT.115 device using standard treated capillaries (NanoTemper Technologies). Fluorescence changes resulting from thermophoresis were recorded using blue channel optics of the instrument ($\lambda_{ex}$ = 470 ± 15 nm, $\lambda_{em}$ = 520 ± 10 nm) for the 30 s period of infrared laser heating at 40% of maximum laser power followed by a 5 s of cooling period. Measurements were performed in a buffer containing 50 mM Tris-HCl, pH 8.0, 100 mM NaCl, 5 mM MgCl$_2$, 1 mM DTT and 0.1% Tween 20. 2'-O-(6-[Fluoresceinyl]aminohexylcarbamoyl)-cyclic diguanosine monophosphate (2'fluo-AHC-c-di-GMP, Biolog, Bremen, Germany) was added to probe the binding affinity of CleD domains. A 16-point 1:1 serial dilution series of proteins was titrated against a fixed concentration (60 nM) of 2'fluo-AHC-c-di-GMP. Data were evaluated in the program ProFit (Quansoft, Zurich, Switzerland) and fitted using the Hill equation.

## Size exclusion chromatography coupled with multi-angle light scattering

The oligomeric state of His-CleD-GFP in the presence or absence of c-di-GMP was determined using a S200 5/150 column (GE Healthcare) mounted to an Agilent 1000 series HPLC (Agilent Technologies). The setup was connected with a miniDwan TriStar multiangle light scattering detector (Wyatt Technology) and Optilab rRex refractive index detector (Wyatt Technology). CleD-GFP (20 µl) was loaded on the column in the absence or presence of c-di-GMP at 5:1 molar ligand/protein ratio. Flow rate was 0.5 ml/min using SEC buffer (see protein expression and purification in the main section). Experiments were performed at 6°C. The mass distribution and molecular weight of samples were determined using the ASTRA five software (Wyatt Technology). The extinction coefficient of c-di-GMP were $\varepsilon_{253}$ = 28,600 M-1 cm$^{-1}$ and $\varepsilon_{280}$ = 17160 M$^{-1}$ cm$^{-1}$ and for CleD-GFP, an $\varepsilon_{280}$ = 31860 M$^{-1}$ cm$^{-1}$ was calculated using EXPASY server.

## DNA work

For amplification of *C. crescentus* genes by polymerase chain reaction (PCR), Phusion polymerase (NEB) with GC buffer was used. Chromosomal deletion constructs were verified by PCR using Taq polymerase (NEB). Classical cloning was performed using the indicated restriction enzymes (NEB) and ligase (NEB). Inserts with base pair (bp) exchanges to create point mutants in the respective proteins were generated by using overlap PCR followed by restriction and ligation. Constructs fused to *gfpmut1* or *sumo* were generated using restriction-free (RF) cloning (*Bond and Naus, 2012*).

## High speed video tracking and analysis

Cells were grown to early exponential phase in PYE. Cells were diluted by fresh PYE to lower cell density and then sealed between a slide glass and a cover slip using VALAP. Phase contrast images of swimming bacteria were observed at 30°C on the inverted scope (Nikon Eclipse TI-U) with 10x objective lens (Nikon CFI Plan Fluor). Time-lapse movies were recorded with a digital scientific CMOS camera (Hamamatsu ORCA-Flash4.0 V2) at 16 frames per second.

Swimming trajectories were reconstructed from time-lapse movies by the algorithm developed by *Dufour et al., 2016*. Motile swarmer cells with trajectory time longer than 10 s were analyzed for

reversal events.For each trajectory, reversal events were detected based on the angular acceleration in the 2D field of view. At this frame rate reversals took place in less than two frames. Defining $\theta_i$ as the angle between consecutive velocity vectors, the criteria for reversal was that the angular acceleration $\Delta\theta_i = \theta_{i+1} - \theta_i$ changes sign over two consecutive frames, $\Delta\theta_i\Delta\theta_{i+1} < 0$, and the magnitudes over the two frames are larger than 45 degree, $|\Delta\theta_i| > 45$, $|\Delta\theta_{i+1}| > 45$. Using visual inspection, we verified this criterion accurately detected the individual reversal events. Swimming velocity was determined by calculating mean velocity of swimming events, that is excluding reversal events. For each strain analysed, data from >850 cells was collected.

## Tethering assay

Cell samples were prepared in the same way as for high-speed video tracking. After dilution, cells were located in Chambered #1.0 Borosilicate Coverglass System (Lab-Tek). Phase images of bacteria that attached to the glass surface were recorded with 100x oil immersion objective (Nikon CFI Plan Fluor) at 50 frames per second.

## Nuclear magnetic resonance (NMR) spectrometry

NMR spectra were recorded at 298 K on Bruker DRX 600, and DRX 900 NMR spectrometers equipped with TXI and TCI probe heads, respectively. The one dimensional NMR spectra were recorded with a Jump-return echo sequence (*Sklenár et al., 1987*) optimized for resonances at around 12 ppm. For the complex His-CleD-GFP·c-di-GMP and apo His-CleD-GFP 16384 scans were recorded. The NMR samples were prepared as follows. The sample of apo His-CleD-GFP (30 µM) was prepared in 100 mM NaCl, 20 mM sodium phosphate buffer at pH 6.5, 0.01% NaN$_3$ (w/v), and 5% (v/v) D$_2$O as sample volume of 250 µL. Complex CleD-GFP·c-di-GMP: the ligand was titrated to the apo His-CleD-GFP sample from a 20 mM stock solution in twofold excess.

The standard two-dimensional spectra $^1$H-$^{15}$N HSQC of His-CleD with or without peptides were performed on a $^{15}$N isotope labeled His-CleD. The His-CleD (400 µM) sample was in 20 mM sodium phosphate buffer at pH 7.0, 200 mM NaCl, 1 mM MgCl$_2$, 5 mM 2-Mercaptoethanol, 0.02% NaN$_3$ (w/v), and 5% (v/v) D$_2$O with a sample volume of 270 µL. c-di-GMP was titrated to the protein sample in 2.5-fold excess. The FliM peptide from residue Ala47 to Gly62 contains an additional Trp at the N-terminus to measure concentration and to increase solubility (WASERILNQDEIDSLLG). As the corresponding ID57_WA peptide contains a Trp we used the following sequence ASERILNQDE-WASLLG. Both peptides were purchased from JPT Peptide Technologies, and dissolved in 200 mM NaCl, 20 mM sodium phosphate buffer at pH 7.0, 1 mM MgCl$_2$, 5 mM β-Mercaptoethanol, 0.02% NaN$_3$ (w/v), and 5% (v/v) D$_2$O. With homonuclear 2D-TOCSY and 2D-NOESY experiments the peptide proton resonances were assigned. It was confirmed, that both peptides were stable in this buffer and unfolded (data not shown). The spectrum of the purified CleD protein was recorded in the same buffer as the FliM peptides.

## Bacterial two hybrid analysis

Bacterial two hybrid screens were performed as published (*Karimova et al., 1998*). Full open reading frames or gene fragments were fused to the 3′ end of the T25 (pKT25), the 3′ end of the T18 (pUT18C) or the 5′ end of the T18 (pUT18) fragment of the gene coding for *Bordetella pertussis* adenylate cyclase. Plasmids were transformed into AB1770 and plated on LB plates containing Kan and Amp. Two microliter of an O/N culture containing the appropriate plasmids were spotted on a MacConkey Agar Base plate supplemented with Kan, Amp and maltose and incubated at 30°C.

## Microfluidics

Polydimethylsiloxane (PDMS) microfluidics devises were produced as described elsewhere (*Deshpande and Pfohl, 2012*). Fresh medium was fed from a 1 ml syringe driven by a pump (Type 871012, B-Braun Melsungen AG) adjusted to carry 1 ml syringes and applying flow rates of 0.002 µm/s in the flow channel. Flow channels were 1 cm in length, 40 µm in width and 15 µm in height. Bacteria were introduced through the outlet, slowly approaching the front of the sterile medium which was kept in place while the air separating the two liquid phases escaped through the PDMS. Medium flux was initiated immediately after the two liquid phases merged. Microchamber devices were designed as rows of microchambers connected to one of two main channels between inlet and

outlet of the same dimensions as described above. Chambers were squares of 40 μm and 0.75 μm in height.

## Chemotaxis assay

This assay was performed basically as described recently (*Briegel et al., 2011*) on M2 (no carbon source added) or P2 (no nitrogen or carbon source added) semi solid agar plates (0.25% agar). Small, round paper discs (Oxoid X3498) were placed on the agar plates and 10 μl of the C- or C- and N-source was dropped on the disc. A colony of NA1000 or mutants was stabbed in the agar around 1 cm away from the disc. The plates were incubated for 4 to 5 days at 30°C until the swarm was visible.

## Bioinformatics

The protein sequences of the *C. crescentus* RRs and *E. coli* CheY were retrieved from Uniprot server. Multiple sequence alignments were performed using the MAFFT program (scoring matrix BLO-SUM62). For the phylogenetic tree construction, 46 C. crescentus RR sequences were aligned as described above and the non-conserved sequences at the N- and C-termini were manually trimmed, preserving the core RR sequences with the sequence numberings corresponding to residues 18–138 and 21–141 for CleA and CleD, respectively. The phylogenetic tree was constructed by PHYML program. All programs mentioned above were performed in Geneious Pro 7.1.7 (Biomatters, Auckland, New Zealand).

The homology models of CleD and CleA were generated in MODELLER using the *E. coli* CheY (PDB 1F4V) as template. The CleA and CleD models were superimposed onto *E. coli* CheY and the *C. crescentus* FliM model was built by mutating the *E. coli* FliM residues in silico using the program PyMOL (The PyMOL Molecular Graphics System, Version 1.7.4 Schrödinger, LLC on World Wide Web http://www.pymol.org). To identify ARR domain proteins in silico, Jackhmmer (*Finn et al., 2011*) and PSI-Blast (*Altschul et al., 1997*) were queried with the ARR domain of CleD (aa 142–173) against the non-redundant database (NR) with default settings and until convergence. Results were merged and resulted in 302 hits. Proteins were then spotted on a simplified version of the genomic based bacterial phylogenic tree provided by Segata et al. (*Segata et al., 2013*). Taxa missing in the genomic tree were added according to 16S rRNA similarity. Sequence logo of the ARR was generated with WebLogo 3.4 (*Crooks et al., 2004*; *Schneider and Stephens, 1990*) based on the multiple sequence alignment generated by Jackhmmer (*Finn et al., 2011*) when queried with the region 142–173 of CleD against the rp75 with default settings with the exception of PAM30 as matrix, until convergence (i.e. seven iterations).

Sequence logos of the rec domain from proteins either with or without an ARR were generated on the Weblogo http://www.twosamplelogo.org/cgi-bin/tsl/tsl.cgi (*Vacic et al., 2006*). The 136 best blastP hits of the CleD rec domain (residues 21–141) found in the rp75 were separated in two groups according to the presence/absence of the ARR domain in the corresponding protein sequences. Both sequence datasets were aligned together against the PFAM (*Sonnhammer et al., 1997*) model corresponding to this region (PF00072) using hmmalign (*Finn et al., 2011*). For the sake of readability, regions of the alignments covered by less than 90% of the sequences were discarded from the logo.

## Acknowledgements

We thank Fabienne Hamburger for cloning and strain constructions, and Benoit-Joseph Laventie and Antje Hempel for help with data analysis. We are grateful for experimental support by Timothy Sharpe (Biophysics Core Facility), Timo Glatter and Alexander Schmidt (Proteomics Core Facility), and Vesna Olivieri (University Microscopy Center). We thank Martin Thanbichler, Christine Jacobs-Wagner, Patrick Viollier, Sebastian Hiller and Dirk Landgraf for providing strains or plasmids. This work was supported by the Swiss National Science Foundation (SNF) Sinergia grant CRSII3_127433 and by an ERC Advanced Research Grant to UJ; by the Paul Allen foundation (award no. 11562) and by the National Institute of Health (grant no. 1R01GM106189) to TE; by the Swiss National Science Foundation (SNF) grant 31003A_166652 to TS; and by the Swiss National Science Foundation (SNF) grant 31003A_173089 to SG.

## Additional information

### Funding

| Funder | Grant reference number | Author |
|---|---|---|
| European Research Council | Advanced Research Grant to U.J. | Urs Jenal |
| Paul G. Allen Family Foundation | 11562 | Thierry Emonet |
| Schweizerischer Nationalfonds zur Förderung der Wissenschaftlichen Forschung | Sinergia grant CRSII3_127433 | Urs Jenal |
| National Institutes of Health | 1R01GM106189 | Thierry Emonet |
| Schweizerischer Nationalfonds zur Förderung der Wissenschaftlichen Forschung | 31003A_166652 | Tilman Schirmer |
| Schweizerischer Nationalfonds zur Förderung der Wissenschaftlichen Forschung | 31003A_173089 | Stephan Grzesiek |

The funders had no role in study design, data collection and interpretation, or the decision to submit the work for publication.

### Author contributions

Jutta Nesper, Conceptualization, Validation, Investigation, Methodology, Writing—original draft, Writing—review and editing; Isabelle Hug, Formal analysis, Validation, Investigation, Methodology, Writing—original draft, Writing—review and editing; Setsu Kato, Judith Maria Habazettl, Formal analysis, Investigation, Methodology, Writing—review and editing; Chee-Seng Hee, Formal analysis, Investigation, Visualization, Methodology, Writing—review and editing; Pablo Manfredi, Formal analysis, Investigation, Writing—review and editing; Stephan Grzesiek, Resources, Funding acquisition, Project administration, Writing—review and editing; Tilman Schirmer, Conceptualization, Resources, Supervision, Funding acquisition, Project administration, Writing—review and editing; Thierry Emonet, Conceptualization, Resources, Formal analysis, Supervision, Funding acquisition, Writing—original draft, Project administration, Writing—review and editing; Urs Jenal, Conceptualization, Resources, Formal analysis, Supervision, Funding acquisition, Investigation, Visualization, Writing—original draft, Writing—review and editing

### Author ORCIDs

Isabelle Hug, http://orcid.org/0000-0002-4524-9569
Thierry Emonet, http://orcid.org/0000-0002-6746-6564
Urs Jenal, http://orcid.org/0000-0002-1637-3376

### Decision letter and Author response

Decision letter https://doi.org/10.7554/eLife.28842.022
Author response https://doi.org/10.7554/eLife.28842.023

## Additional files

### Supplementary files

• Supplementary file 1. (**A**) Strains used in this study. (**B**) Plasmids used in this study. (**C**) Oligonucleotids used in this study.
DOI: https://doi.org/10.7554/eLife.28842.020

• Transparent reporting form
DOI: https://doi.org/10.7554/eLife.28842.021

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
