## [Decision Letter]

Thank you for submitting your article "Cyclic di-GMP differentially tunes a bacterial flagellar motor through a novel class of CheY-like regulators" for consideration by *eLife*. Your article has been reviewed by two peer reviewers, and the evaluation has been overseen by a Reviewing Editor and Gisela Storz as the Senior Editor. The following individual involved in review of your submission has agreed to reveal her identity: Fitnat H Yildiz (Reviewer #3).

The reviewers have discussed the reviews with one another and the Reviewing Editor has drafted this decision to help you prepare a revised submission.

In this manuscript, Nesper et al. describe results from studies designed to identify and characterize novel c-di-GMP effectors in *Caulobacter crescentus*.

The manuscript can be divided in two sections:

In the first part, clear and complete, the authors carry out extensive characterization of Cle proteins, measuring c-di-GMP binding affinity and stoichiometry, and showing that the Arg-rich region (ARR) in all the Cle proteins is necessary and sufficient to bind c-di-GMP, with one more conserved R in the Rec domain contributing to binding. Motifs with Arg residues have been identified earlier and are present in every active site of solved effector proteins structures (PilZ, PgaC, Bld), where they are thought to play an important role in the stabilization of bound c-di-GMP. The ARR region in the Cle proteins identified in this study has a different motif. Like the first-identified PilZ effector, CleD is also monomer that binds a dimeric c-di-GMP. Using ARR sequence as a query, the authors identified over 300 proteins in other bacterial phyla, the majority being CheY homologs, suggesting perhaps a specific function for this motif in modulating flagellar motor behavior/chemotaxis.

The second part, on the other hand, is much more difficult to follow and digesting the massive amounts of data from experiments that often appear unconnected, is difficult. In general, multiple motility and adhesion assays that don't correlate easily are presented and the use of different Cle proteins for different assays makes the story confusing. Although it appears that some Cle proteins have a role in connecting Flagellum and adhesion via the stalk, the exact connection is not clear. Perhaps, the clearest result to emerge was that only one of five Cle proteins (CleA) controls flagellar motor reversals and hence chemotaxis, as well as motor speed. CleA apparently has an antagonistic action to that of the main chemotaxis regulator CheYII, and is thus likely to 'tune' motor behavior in response to c-di-GMP. The motor tuning mechanism observed for CleA is similar to other c-di-GMP effectors known to control chemotaxis in other bacteria at various points: YcgR, a PilZ-like c-di-GMP effector controls motor reversals and motor speed via interaction with the flagellar switch proteins FliM/FliG as well as with stators in *E. coli/Salmonella*. Thus, the CleA data presented constitute a variation on established themes.

In sum, for a broad audience, it is essential that the authors do a profound rewrite of the second part and clarify data presentation. Critically, the authors must resolve conflicting observations and conclusions (see below). Overall, the results show that Cle protein form a new family of CheY-like c-di-GMP effectors that regulate the flagellum directly in multiple but not redundant manner. The exact details of these functions are still missing.

Specific comments

1) In a previous work, the Jenal group undertook an extensive study of *C. crescentus* proteins with predicted diguanylate cyclase (DGC) and/or phosphodiesterase (PDE) activities (PLoS Genet 9(69)-e100374). It would be critical to determine if there is an interaction between these DGCs and the Cle system. For example, is the DGC DgcB, reported to be the main DGC dedicated to motility regulation, required for Cle function? Addressing this could position the Cle proteins in the DGC pathway.

2) Is there any information on the expression of Cle proteins? Is CleB likely to contribute to any of the phenotypes discussed?

3) Abstract: “We identified a novel family of CheY-like (Cle) proteins in *Caulobacter crescentus*, which tune flagellar activity in response to binding of the second messenger c-di-GMP to a short C-terminal extension.” This statement is not correct. Only one member of this family was shown to affect flagellar activity.

“In their c-di-GMP bound conformation Cle proteins interact with the flagellar switch to control motor activity.” This statement is also not quite correct. The requirement for c-di-GMP binding for promoting FliM interaction was implicated (by mutation of the R residues) for 2/5 proteins (CleA and CleD) in the data shown in Figure 5. In Figure 6, it does not appear the interaction between CleA/D and FliM was tested in the presence and absence of c-di-GMP in any of the three assays shown.

4) There are major concerns linked to Figure 4.

Motility. What accounts for the different motility behavior of single deletions of Cle C, D, and E in (A) vs (B) where levels of c-di-GMP are different? In fact, one learns later (in Figure 7) that the CleA mutant, which does nothing in (A) and (B) has a hyper-reversal behavior when monitored by other assays. In (B), where only the single deletion of CleD increases motility somewhat, why does deletion of all 5 increase motility over the WT? The PYE motility medium should reflect both speed and chemotaxis. Since only CleA affects either parameter, why are the multiple deletions doing so much better in (A)?

Adhesion. The other difficult aspect of assessing data in this figure is that when first introduced in the text, the adhesion assays are not discussed, and the reader has to wait until much later after the data in Figure 7 is presented. Like with the motility assays, the three different adhesion assays are also confusing. For example, why does deletion of CleC, D or E have no effect in the adhesion assay in (C) but has an effect in (E)? How is the polystyrene assay in (c) different from the holdfast assay in (E)?

Subsection “CleC and CleE promote surface recognition and rapid attachment”: “A mutant lacking all five Cle proteins showed severely reduced attachment to polystyrene (Figure 4)”. Is a 50% reduction in polystyrene binding really 'severe'? If so some comparisons are not clear: In Figure 4 50% reduction in attachment of mutants lacking CleC, D or E, as well as the CheYII mutant is observed. Yet, the former set is characterized as “severely impaired” and the latter as “not impaired”. The conclusion 'that interaction of one or several Cle proteins with the motor is required for the ability to rapidly synthesize an adhesive holdfast in response to surface exposure is not justified.

In (D) two cells are shown – ST cell that gives rise to the SW cell, which will then become an ST cell. Is the adhesion of both being monitored in (E)? Don't they have different c-di-GMP levels? Will that not impact the interpretation?

MotA or MotB data are missing in Figure 4.

“These results also imply that the role of Cle proteins in rapid surface attachment is mediated by their c-di-GMP dependent interaction with the motor switch.” Again this conclusion is questionable because Figure 4 merely shows that PilA mutants and all-Cle mutants each have a 3 minute delay in holdfast appearance, while a combination of these mutations has a 25 min delay i.e. no regulation. Neither of these observations implicate the motor. Perhaps CleB-E, which don't have a discernible effect on motor reversal (hence FliM binding), instead regulate the pilus?

In general, statistics are missing in this figure and other figures to test the significance of some effects (i.e. is the cleD mutant distinct from WT in Figure 4?)

5) Figure 5. Is it really necessary to test localization using so many variants of two different proteins? ARR was already established as binding c-di-GMP. Why were the localization experiments not done with *ΔpdeA* i.e. at elevated levels of c-di-GMP?

Figure 5—figure supplement 1 shows localization of CleC, but this is not mentioned in the text. If CleB and E do not localize to the base of the flagellum, how does one interpret the effects of the various effects of deletions encompassing these two proteins in the assays shown in Figure 4?

6) Figure 6. CleD pulls down FliM, as well as shows binding to a FliM peptide by NMR. Is this interaction dependent on c-di-GMP. Similarly for the two-hybrid assay showing CleA-FliM interaction. Why not test the other Cle proteins in this assay?

7) Figure 7. It is only when we get to this figure showing that CleA affects motor reversals, that it becomes clear why CleA was being used in the previous experiments (in addition to the soluble CleD). Again, however, the motility assays in Figure 4 were indicating a role for Cle-CDE, not CleA. How is this resolved? In light of the fact that the Arg domains in all the Cle proteins bind c-di-GMP, but not all showed robust localization or flagellar reversal modulation, why were reversal frequencies and motor speeds of the various Cle mutants not measured in the *ΔpdeA* background?

Figure 7—figure supplement 2. Why is the assay shown in Figure 4 not a chemotaxis assay? Bacteria will consume nutrients as they grow and generate a gradient; tryptone plates are classically used to study chemotaxis. The chemotaxis effects shown in this figure are poorly visible and not quantified.

[Editors' note: further revisions were requested prior to acceptance, as described below.]

Thank you for resubmitting your article "Cyclic di-GMP differentially tunes a bacterial flagellar motor through a novel class of CheY-like regulators" for consideration by *eLife*. The evaluation has been overseen by a Reviewing Editor and Gisela Storz as the Senior Editor.

The editors have discussed the changes made to the original submission with one another and the Reviewing Editor has drafted this decision to help you prepare a revised submission.

In this revised version of the manuscript, the authors have made substantial efforts to address the scientific concerns raised by the reviewers, adding additional experiments and controls where needed. The editors are convinced that this new data addresses the reviewer's concerns from a purely scientific point of view.

However, the manuscript remains a difficult read, largely because the number of specific assays, genetic backgrounds and combination make it very hard to derive a clear picture of the Cle protein network. In fact, the functions of the Cle proteins appear to be diverse and are only partially established by the study. Demonstrating the exact function of each Cle protein at the molecular level is probably beyond the scope of the manuscript; but trying to address the function of a many as five Cle proteins in up to eight multi panel figures is overwhelming, and affects the potency (and thus significance) of the manuscript.

To improve the readability and impact of the paper, the structure of the manuscript should be greatly simplified to highlight the major finding that Cle proteins form a new class of response regulators that may not necessitate phosphorylation but bind ci-diGMP to regulate flagellar function. The in vivo data should be restricted to a clear example, perhaps the function of CleA in chemotaxis, while the other possible functions should be saved for the discussion, highlighting the potential functional diversity of these proteins and connection to swimming and adhesion.

An aside with respect to Figure 4—figure supplement 1 and in Figure 8—figure supplement 1, because these figures are not central to paper and because improving the quality will be hard, we think the best solution is probably to remove them.

---

## [Author Response]

[…] In sum, for a broad audience, it is essential that the authors do a profound rewrite of the second part and clarify data presentation. Critically, the authors must resolve conflicting observations and conclusions (see below). Overall, the results show that Cle protein form a new family of CheY-like c-di-GMP effectors that regulate the flagellum directly in multiple but not redundant manner. The exact details of these functions are still missing.

The manuscript was restructured and streamlined in order to make this part clearer and more accessible for a broad readership. We now integrate our results with the findings of Hug et al. and carefully explain the different motility and adhesion assays to clarify how these experiments are connected and how they contribute to the functional characterization of the Cle proteins.

We believe that part of the confusion has arisen from the fact that the general “motility” assay used in Figure 4 (Figure 6 in the revised version, see below) and 8D does not represent a genuine motility readout, but rather measures spreading of bacteria on semisolid agar plates. Although this assay is often used to quantify swimming or chemotaxis, spreading is in fact a complex combination of growth, motility, chemotaxis and surface (agar) adherence. Thus, when referring to this assay we replaced the terms “swimming” or “motility” with the more neutral term “spreading”. This also refers to the labeling of the Y-axis in Figure 6 (formerly Figure 4) and 8D. Finally, we have changed the title of this particular chapter from “Cle proteins modulate *C. crescentus* motility” to “Cle proteins modulate *C. crescentus* spreading on semisolid agar plates”

In the revised version, we also make it clear to the reader that to follow up on the semisolid agar experiments more specific assays were used to further dissect the role of individual Cle proteins in any of these behavioral aspects (motility, chemotaxis, attachment), including single cell tracking for changes in speed or reversal rates and microfluidics for rapid surface adherence (as introduced by Hug et al., 2017). We are now emphasizing the differences of the individual assays in the text (see below). E.g. “Because the ability of cells to spread on semisolid agar mirrors a combination of different forms of behavior including motility, chemotaxis and surface adherence, the specific role of each Cle member in this process remained unclear.”

Finally, we realized that the order of the chapters (and figures) in the original version was not helpful for the understanding of the in vivo results. In particular, introducing the semisolid agar data (formerly Figure 4) and then switching to Cle-motor interaction data only to finally switch back to motility analysis, is indeed somewhat confusing. To make the flow of the story more coherent and logic, we have switched the order of Figure 4, Figure 5, and 6 (with the old Figure 4 now being Figure 6; old Figure 5 now Figure 4; and old Figure 6 now Figure 5). This creates a more logic flow in the Results section, where the biochemistry experiments are now followed by the experiments demonstrating CleA/D-motor interaction and finally by experiments addressing the in vivo function of individual Cle proteins.

In sum, the principle finding of the in vivo part of the manuscript is that different Cle proteins are involved in distinct aspects of moving and adhering. Although the details of this control remain unclear, we propose that the primary role of the Cle proteins is to coordinate flagellar function with physical stimuli during surface encounter thereby influencing the motile/sessile decision making. Thus, while CheY proteins tune the flagellar motor in a changing chemical landscape, we postulate that the Cle proteins integrate physical information.

Specific comments1) In a previous work, the Jenal group undertook an extensive study of C. crescentus proteins with predicted diguanylate cyclase (DGC) and/or phosphodiesterase (PDE) activities (PLoS Genet 9(69)-e100374). It would be critical to determine if there is an interaction between these DGCs and the Cle system. For example, is the DGC DgcB, reported to be the main DGC dedicated to motility regulation, required for Cle function?Addressing this could position the Cle proteins in the DGC pathway.

Hug et al., 2017 showed that the diguanylate cyclase DgcB acts in a pathway with the flagellar motor to increase c-di-GMP levels in response to mechanical stimuli during *C. crescentus* surface contact. Based on the observation that a *ΔcleA-E* mutant phenocopies a D*dgcB* mutant (increased spreading on semisolid agar, strongly reduced ability to rapidly form holdfast and attach to surfaces) and based on the observation that Cle proteins are activated by c-di-GMP, we speculate that DgcB is positioned upstream of one or several of the Cle proteins. We have introduced additional data demonstrating that DgcB is indeed upstream of the Cle proteins in Figure 6—figure supplement 1A, B.

It is possible that additional diguanylate cyclases and phosphodiesterases are involved in regulating the activity of individual Cle proteins. However, since the manuscript is already heavy on data, covering an additional upstream level of the Cle components would overload the story completely.

2) Is there any information on the expression of Cle proteins? Is CleB likely to contribute to any of the phenotypes discussed?

Please note the following statement in the manuscript:

“All five *cle* genes are expressed in PD cells, coincident with the transcription of flagellar and chemotaxis genes [Schrader et al., 2014, Zhou et al., 2015].”

We were not able to find a phenotype or behavioral change for the *ΔcleB* mutant. We assume that this bears on the limited functional space covered by our assays.

3) Abstract:“We identified a novel family of CheY-like (Cle) proteins in Caulobacter crescentus, which tune flagellar activity in response to binding of the second messenger c-di-GMP to a short C-terminal extension.” This statement is not correct. Only one member of this family was shown to affect flagellar activity.

We agree with the reviewer that the data arguing for functional interference with the motor are strongest for CleA. To strengthen the idea that the other Cle proteins also interfere with the flagellar motor, we have added the following experimental data to the revised manuscript:

1) Overexpression of all individual *cle* genes lead to reduced spreading of *C. crescentus* in semisolid agar. Effective interference by *cleC, cleD* and *cleE* was independent of pili but required a functional copy of *dgcB*, arguing that these Cle proteins are downstream components of the DgcB-mediated surface response (see: Hug et al., 2017). The observation that the spreading of *cle* overexpression is independent of pili, indirectly argued for an involvement of the flagellum. These data have been added as supplement 1 of Figure 6 (formerly Figure 4).

2) Rapid attachment of a *ΔcleA-E* mutant is restored by expressing *cleC, cleD* or *cleE* from a plasmid *in trans*. Importantly, this effect was only observed in strains expressing a functional *fliM* allele, while attachment was not restored in a strain expressing the mutant *fliMID57WA* allele, the product of which lacks the binding site for Cle proteins (data added to Figure 6). This strongly argued that CleC, CleD and CleE contribute to rapid surface attachment via their interaction with the flagellum.

3) We have added localization data for CleC to Figure 4, demonstrating that CleC, similar to CleA and CleD, localizes to the flagellated pole in a FliM-dependent manner.

“In their c-di-GMP bound conformation Cle proteins interact with the flagellar switch to control motor activity.” This statement is also not quite correct. The requirement for c-di-GMP binding for promoting FliM interaction was implicated (by mutation of the R residues) for 2/5 proteins (CleA and CleD) in the data shown in Figure 5. In Figure 6, it does not appear the interaction between CleA/D and FliM was tested in the presence and absence of c-di-GMP in any of the three assays shown.

The referee is correct, in that this conclusion was primarily based on indirect data (cell biology, biochemistry) with direct evidence arguing for c-di-GMP mediated interaction of Cle proteins with FliM being missing in the original version of the manuscript. Since CleD is the only member of the Cle family amenable to biochemical studies, we had shown by NMR spectroscopy that full-length CleD interacts with the FliM wild-type peptide but not with the FliM_ID57WA mutant peptide Figure 5 (formerly Figure 6) To strengthen this point and to gather direct evidence for a role of c-di-GMP in motor interaction, we have repeated NMR experiments with full-length CleD and FliM wild-type peptide in the absence of c-di-GMP. The spectrum recorded without c-di-GMP was indistinguishable from spectra with CleD alone or with CleD, c-di-GMP and the FliM_ID57WA mutant peptide. Thus, an intact N-terminal peptide of FliM together with c-di-GMP is required for the interaction observed by NMR.

We have added reference to this experiment in the manuscript text:

“Adding the FliM wild-type peptide in two-fold excess resulted in shift changes of CleD of approximately 10 resonances. In contrast, the addition of the FliM peptide containing the ID57WA substitutions gave no observable chemical shift (Figure 5). Likewise, the addition of the FliM wild-type peptide without c-di-GMP gave no observable shift.”

and in the legend to Figure 5:

“The spectrum of CleD with c-di-GMP and the FliM_ID57WA mutant peptide or with CleD and the wild-type FliM peptide was indistinguishable from the spectrum recorded for CleD alone.”

4) There are major concerns linked to Figure 4.Motility. What accounts for the different motility behavior of single deletions of Cle C, D, and E in (A) vs (B) where levels of c-di-GMP are different? In fact, one learns later (in Figure 7) that the CleA mutant, which does nothing in (A) and (B) has a hyper-reversal behavior when monitored by other assays. In (B), where only the single deletion of CleD increases motility somewhat, why does deletion of all 5 increase motility over the WT? The PYE motility medium should reflect both speed and chemotaxis. Since only CleA affects either parameter, why are the multiple deletions doing so much better in (A)?

General point:

We realized that one reason for the apparent confusion relating to the motility and attachment data in Figure 6 (formerly Figure 4) refers to the term “motility”, which was used throughout the manuscript to describe these experiments. The semisolid agar plate assay measures a combination of several processes like growth, chemotaxis, cell motility, and surface adherence. As indicated above we are now referring to “spreading on semisolid agar” as opposed to “motility” when discussing these data. Also, we have introduced the following sentences to make it clear to the reader how the data in Figure 6 relate to the attachment data:

“We thus hypothesized that some of the Cle proteins mediate physiological changes sparked by (agar) surface contact. This idea is based on the premise that cells moving through semisolid agar will be constantly challenged by (agar) surface and thus may respond to physical cues by reducing their motility and boosting their propensity to attach.”

Specific points:

As outlined in the manuscript it is indeed counterintuitive that a *ΔcleA* mutant, which in a homogenous environment (quasi 2-D chamber, Figure 7) shows a hyperreversal phenotype, is completely unaffected in its ability to spread on semisolid agar (Figure 6; formerly Figure 4). We argued in the manuscript that CleA and CheYI compete with CheYII for motor binding and that “the strongly increased reversal rates observed for these mutants in a homogenous environment are adjusted in chemical gradients, possibly by adapting the activity of CheYII or additional functional CheY homologs.”

The referee is rightly pointing out that individual *cle* deletion mutants show different phenotypes in the assays outlined in Figure 6 (formerly Figure 4). It is important to note that the genetic background of the assay strains in these two experiments is different. While experiments in 6a were carried out in a c-di-GMP wild-type background, experiments in 6b were done in a strain lacking PdeA, which has increased levels of c-diGMP and, as a consequence, shows poor spreading on semisolid agar (Abel et al. 2011 Mol. Cell). Thus, Cle proteins are likely more active in *ΔpdeA* mutant as compared to a *pdeA*^+^ background.

The observation that a *ΔcleD* mutant is the only single *cle* mutant that partially restored spreading in the *ΔpdeA* background indicates that in cells with increased c-di-GMP levels this protein makes the largest contribution towards sessility. However, it is important to note that the other Cle proteins also contribute to the observed anti-spreading phenomenon, since a mutant lacking all five *cle* genes showed much improved spreading as compared to the *ΔcleD* single mutant. It is certainly not unusual for biological processes that the phenotype of specific mutations is dependent on the genetic context.

Finally, why does the *ΔpdeA* D5 mutant lift the degree of spreading to a level that is above wild-type (Figure 6)? Please note that deleting all five *cle* genes improves spreading in agar by a factor of 4 both in the wild-type background (Figure 6) and in the *ΔpdeA* background (Figure 6). And related to this: why are the multiple *cle* deletions doing so much better in the assay in Figure 6 (wt background) as compared to 6B (*ΔpdeA* background)? This is because the c-di-GMP concentration is different. Or, in other words, our data argue that spreading in semisolid agar is inhibited by c-di-GMP but not only via the Cle proteins but also via some other, unknown pathways. If one would delete all factors blocking spreading at high c-di-GMP one would expect that the overall spreading in Figure 6 would be similar, irrespective of their c-di-GMP background/level.

Adhesion. The other difficult aspect of assessing data in this figure is that when first introduced in the text, the adhesion assays are not discussed, and the reader has to wait until much later after the data in Figure 7 is presented. Like with the motility assays, the three different adhesion assays are also confusing. For example, why does deletion of CleC, D or E have no effect in the adhesion assay in (C) but has an effect in (E)? How is the polystyrene assay in (C) different from the holdfast assay in (E)?

Whereas a generic 24 hrs attachment assay to polystyrene surface was used in Figure 6 (formerly Figure 4), experiments in Figure 6/E and F/G make use of more sophisticated microfluidic-based attachment assays that were pioneered by Hug et al. 2017 and allow measuring surface sensing and attachment of newborn swarmer cells within a timescale of seconds to minutes. For instance, in the assay shown in Figure 6/E strong flow removes new offspring immediately after separating from their mothers, unless they are able to sense surface and produce an adhesive holdfast. Similarly, the assay in Figure 6/G uses very small chambers in which newborn swarmer cells are constantly challenged by surface during and after division, thus allowing to determine the minimal time required for holdfast synthesis in response to physical cues from the surrounding surface. The observed differences in behavior of individual *cle* mutants (e.g. Figure 6) bear on these fundamental experimental differences and assay conditions. For instance, attachment to plastic after 24 hours is a relatively crude assay that can provide only very general information about a process that occurs on a timescale of minutes. Phenotypes that are discernible in microfluidic-based assays may be easily masked in a long-term assay like the one shown in Figure 6. The fact that deletion of *cle* genes strongly affected rapid attachment under flow conditions but showed a more moderate defect in a 24h attachment assay, argues for a specific role of Cle proteins in mechanosensation and short-term attachment.

The revised version of the manuscript introduces the individual attachment assays and explains their specific differences. E.g. we now state:

“To test if the observed attachment defect of cle mutants results from a reduced capability to sense and respond to surface interactions, we used microfluidic-based assays that allow to directly score surface attachment of SW cells [Persat, Stone and Gitai, 2014]. When dividing cells are cultured under strong medium flow, the newborn swarmer progeny rapidly deploy an adhesive holdfast upon surface contact and thus are able to adhere before being washed out [Hoffman et al., 2015, Hug et al., 2017] (Figure 6).”

and also:

“Finally, we made use of microfluidic devices with small, quasi-2D side chambers (0.75 µm in height) to enhance surface encounter of newborn swarmer cells [Hug et al., 2017]. This assay allows following SW progeny microscopically from the moment of birth at division to the onset of holdfast production (Figure 6). The tight geometry of such microchambers offers cells constant surface interaction opportunities without medium flow.”

Also, we omitted the following sentence that had made reference to attachment data prematurely in the chapter on motility:

“This effect was specific for motility since deleting cleD failed to re-establish normal surface attachment of the *C. crescentus* pdeA mutant (see below, Figure 6—figure supplement 1A).”

In the revised version of the manuscript, all attachment data are now merged in the chapter “CleC and CleE promote surface recognition and attachment”.

Subsection “CleC and CleE promote surface recognition and rapid attachment”: “a mutant lacking all five Cle proteins showed severely reduced attachment to polystyrene (Figure 4)”. Is a 50% reduction in polystyrene binding really 'severe'? If so some comparisons are not clear: In Figure 4 50% reduction in attachment of mutants lacking CleC, D or E, as well as the CheYII mutant is observed. Yet, the former set is characterized as “severely impaired” and the latter as “not impaired”. The conclusion 'that interaction of one or several Cle proteins with the motor is required for the ability to rapidly synthesize an adhesive holdfast in response to surface exposure is not justified.

The referee correctly points to a few cases where the wording of our manuscript was ambiguous or misleading. Thank you.

We have rephrased the sentence to:

“In agreement with this idea, mutants lacking one or several Cle proteins showed reduced attachment to polystyrene during a 24 hrs attachment assay (Figure 6).”

The referee is correct, in that the *ΔcheYII* mutant does show reduced attachment in the flow chamber assay similar to some of the *cle* single mutants (Figure 6). However, the reduction is considerably stronger for *ΔcleA-E* and for the *ΔcleCE* mutants. This, together with the observation that expression of *cleC, cleD* or *cleE* alone can restore rapid surface attachment of the *ΔcleA-E* strain (new data added to Figure 6), argued that CleC and CleE contribute significantly to this process.

The following sentence:

“Similarly, mutants lacking all Cle proteins or mutants lacking CleC, CleD or CleE showed severely impaired attachment” was replaced with:

“Similarly, mutants lacking all Cle proteins or mutants lacking CleC and CleE showed severely impaired attachment.”

In (D) two cells are shown – ST cell that gives rise to the SW cell, which will then become an ST cell. Is the adhesion of both being monitored in (E)? Don't they have different c-di-GMP levels? Will that not impact the interpretation?

In this microfluidics-based assay SW offspring of attached mother cells are exposed to surface because the medium flow pushes crescentoid cells down towards the surface and positions the flagellated and piliated pole in close proximity to the substratum (see: Persat et al. 2014. Nature Com 5, 3824). This results in rapid holdfast biogenesis at the flagellated pole (preceding the developmental program by 10-20 minutes) and adherence of the daughters to the surface. *Caulobacter* cells attach only once, as holdfast glues them irreversibly to surfaces. Therefore, this assay only monitors the attachment efficiency of newborn swarmer offspring. This is dependent on the capacity to sense and respond to the mechanical trigger of surface contact.

To better explain this, we have added information in the Results section describing the system and referring to the work by Hug et al., 2017. We have also added a detailed description of the Methodology used in the Materials and methods.

MotA or MotB data are missing in Figure 4.

We would like to thank the referee for raising our attention to this issue. This has now been fixed in the text and with the *motB* mutants being incorporated as controls in Figure 6.

“These results also imply that the role of Cle proteins in rapid surface attachment is mediated by their c-di-GMP dependent interaction with the motor switch.” Again this conclusion is questionable because Figure 4 merely shows that PilA mutants and all-Cle mutants each have a 3 minute delay in holdfast appearance, while a combination of these mutations has a 25 min delay i.e. no regulation. Neither of these observations implicate the motor. Perhaps CleB-E, which don't have a discernible effect on motor reversal (hence FliM binding), instead regulate the pilus?

The entire sentence says: “Together with the data shown above, these results also imply that the role of Cle proteins in rapid surface attachment is mediated by their c-di-GMP dependent interaction with the motor switch.”

Thus, the conclusion drawn in this statement combines the in vivo observations from Figure 6 (formerly Figure 4) with experiments demonstrating that one of the Cle proteins, CleD, interacts with the motor in a c-di-GMP dependent manner (although this is the only member of this family that was amenable to biochemistry, we presume that the other Cle proteins behave similarly) and that a motor mutant that is unable to bind Cle proteins (FliM_ID57WA_) shows a similar attachment phenotype like the strain lacking all five Cle mutants.

To tone this statement down and to stress the speculative character of this sentence, we have rephrased this part to:

“Together with the data shown above, these results suggest that the role of Cle proteins in rapid surface attachment may be mediated by their c-di-GMP dependent interaction with the motor switch.”

However, we have added additional experimental evidence to the manuscript arguing against Cle proteins acting through the *C. crescentus* pilus (see also comments to point 3, above).

1) Strains containing single or multiple *cle* deletions showed improved spreading on semisolid agar both in a pilus^+^ and pilus^-^ background (data added to Figure 6—figure supplement 1A).

2) Conversely, overexpression of *cleC, cleD* or *cleE* lead to a strong reduction in cell spreading in semisolid agar both in a pilus^+^ and pilus^-^ background (data added to Figure supplement 1B).

3) Rapid attachment of a *ΔcleA-E* is (partially) restored by expressing *cleC, cleD* or *cleE* from a plasmid in trans. Importantly, this effect was only observed in strains expressing a functional *fliM* allele, while attachment was not restored in a strain expressing the mutant *fliMID57WA* allele, the product of which lacks the binding site for Cle proteins (data added to Figure 6).

4) Deletion of *cleA-E* together with *pilA* exacerbated the delay of holdfast appearance in quasi 2D chambers compared to strains lacking only Cle proteins or pili. This indicated that Cle proteins and pili are not in the same pathway.

In general, statistics are missing in this figure and other figures to test the significance of some effects (i.e. is the cleD mutant distinct from WT in Figure 4?)

Statistical specifications have been added to the legend of Figure 6 (formerly Figure 4).

5) Figure 5. Is it really necessary to test localization using so many variants of two different proteins? ARR was already established as binding c-di-GMP. Why were the localization experiments not done with ΔpdeA i.e. at elevated levels of c-di-GMP?

The localization data shown in Figure 4 (formerly Figure 5) not only complement and strengthen the FliM-CleA/D interaction data but they also serve to probe the requirement of specific features of these proteins for their activity. However, we agree with the referee that the figure is a bit heavy and we have omitted two strains in Figure 4, namely the D*fliF* and the D*fliG* strains. These were meant in support of a functional flagellar basal body structure being required for the localization of CleA and CleD. However, this claim is already covered by the D*fliM* mutant.

We have tested localization of the Cle proteins in a *ΔpdeA* strain and found the same behavior as in the wildtype background.

Figure 5—figure supplement 1 shows localization of CleC, but this is not mentioned in the text. If CleB and E do not localize to the base of the flagellum, how does one interpret the effects of the various effects of deletions encompassing these two proteins in the assays shown in Figure 4?

We have added CleC localization data to Figure 4 (formerly Figure 5).

In contrast to CleA, CleC and CleD, we were not able to show polar localization of CleB and CleE with the assay conditions used for these experiments.

A *cleB* mutant did not show a phenotype in any of the assays shown in Figure 6. In contrast, genetic evidence indicated that CleE is required for efficient surface attachment. A failure to observe localization of this protein in our assays does not necessarily mean that it does not localize to the flagellar base under specific conditions, e.g. conditions favoring surface recognition more strongly than on agar pads, which were used for localization experiments. Also, it is possible that the CleE fusion to the fluorescent reporter is simply not stable.

6) Figure 6. CleD pulls down FliM, as well as shows binding to a FliM peptide by NMR. Is this interaction dependent on c-di-GMP. Similarly for the two-hybrid assay showing CleA-FliM interaction. Why not test the other Cle proteins in this assay?

We have tried to show FliM interaction with CleB, CleC and CleE. Unfortunately, so far, these attempts were not successful.

7) Figure 7. It is only when we get to this figure showing that CleA affects motor reversals, that it becomes clear why CleA was being used in the previous experiments (in addition to the soluble CleD). Again, however, the motility assays in Figure 4 were indicating a role for Cle-CDE, not CleA. How is this resolved?

See comments to point 4 above.

In light of the fact that the Arg domains in all the Cle proteins bind c-di-GMP, but not all showed robust localization or flagellar reversal modulation, why were reversal frequencies and motor speeds of the various Cle mutants not measured in the ΔpdeA background?

We found that the fraction of cells with localized CleA, CleC, and CleD were similar in wild type and the *ΔpdeA* mutant.

Analyzing Cle protein function in strains with different c-di-GMP levels is indeed a great idea. However, we believe that this is beyond the scope of this manuscript.

Figure 7—figure supplement 2. Why is the assay shown in Figure 4 not a chemotaxis assay? Bacteria will consume nutrients as they grow and generate a gradient; tryptone plates are classically used to study chemotaxis. The chemotaxis effects shown in this figure are poorly visible and not quantified.

As outlined above, the assay shown in Figure 4 (now Figure 6) gauges a combination of cell motility, chemotaxis, attachment and growth.

We agree with the referee that the data in Figure 7—figure supplement 2 are poorly visible. Because this figure does not contribute essential information we decided to omit this figure and the associated text from the manuscript.

[Editors' note: further revisions were requested prior to acceptance, as described below.]

*[…] To improve the readability and impact of the paper, the structure of the manuscript should be greatly simplified to highlight the major finding that Cle proteins form a new class of response regulators that may not necessitate phosphorylation but bind ci-diGMP to regulate flagellar function. The* in vivo data should be restricted to a clear example, perhaps the function of CleA in chemotaxis, while the other possible functions should be saved for the discussion, highlighting the potential functional diversity of these proteins and connection to swimming and adhesion.An aside with respect to Figure 4—figure supplement 1 and in Figure 8—figure supplement 1, because these figures are not central to paper and because improving the quality will be hard, we think the best solution is probably to remove them.

We have shortened the manuscript from nine to seven figures and have simplified several of the main figures substantially by removing data, as suggested by the editors.

We have also simplified the manuscript text by condensing the chapters describing the in vivo role of the Cle proteins. We now focus on the role of CleA in chemotaxis (Subsection “CleA interferes with the *C. crescentus* chemotaxis response”) and on the role of Cle in surface attachment (Subsection “Cle proteins promote rapid surface attachment”). The chapter “Cle proteins modulate *C. crescentus* spreading on semisolid agar plates” was omitted. Altogether, these changes reduce the overall length of the manuscript text by 10%.

The Results section is now streamlined with the first three chapters outlining the biochemical and the last four chapters summarizing the functional characterization of Cle proteins:

1) Identification of a CheY subfamily as novel c-di-GMP effectors (Figure 1).

2) An Arg-rich peptide is required and sufficient for c-di-GMP binding of Cle proteins (Figure 2).

3) The conserved ARR c-di-GMP binding motif defines a novel class of CheY proteins (Figure 3).

4) Cle proteins interact with the flagellar basal body upon c-di-GMP binding (Figure 4).

5) CleA interferes with the *C. crescentus* chemotaxis response (Figure 5).

6) Cle proteins promote rapid surface attachment (Figure 6).

7) Cle activation requires c-di-GMP binding but not phosphorylation (Figure 7).